# DO GRAPH NEURAL NETWORK STATES CONTAIN GRAPH PROPERTIES?

## ABSTRACT

Deep neural networks (DNNs) achieve state-of-the-art performance on many tasks, but this often requires increasingly larger model sizes, which in turn leads to more complex internal representations. Explainability techniques (XAI) have made remarkable progress in the interpretability of ML models. However, the non-relational nature of Graph neural networks (GNNs) make it difficult to reuse already existing XAI methods. While other works have focused on instance-based explanation methods for GNNs, very few have investigated model-based methods and, to our knowledge, none have tried to probe the embedding of the GNNs for well-known structural graph properties. In this paper we present a model agnostic explainability pipeline for GNNs employing diagnostic classifiers. This pipeline aims to probe and interpret the learned representations in GNNs across various architectures and datasets, refining our understanding and trust in these models.

## 1 INTRODUCTION

Graph Neural Networks (GNNs) are pivotal in harnessing non-Euclidean graph-structured data (Kipf & Welling, 2017) for tasks ranging from social network analysis to bioinformatics. Despite their success, the black-box nature of GNNs poses significant challenges as classical XAI methods can't be directly applied on GNNs due to the lack of a regular structure (e.g. vertices can have different degrees). In this case, explaining a prediction means identifying important parts of the relational structure, or input features of nodes. An issue is that finding the explanation is itself a combinatorial problem, making XAI methods for GNN intractable (Longa et al., 2023a; Ying et al., 2019).

Previous surveys (Agarwal et al., 2023; Dai et al., 2022) highlighted the lack of comprehensive, robust and model-agnostic explainability methods. We also identified (sec C) few model-level explainability methods. They focus on explaining the decision-making process at a high level, often by generating graph patterns or motifs that influence the model's predictions but none allow for the interpretability of intermediate layers and none highlight the role of graph properties. One paper identified the room for probing classifier (Akhondzadeh et al., 2023), as developed for Natural Language Processing (Giulianelli et al., 2018), (Belinkov, 2021). Their research question was "will the hidden representation contain information about the number of hydrogen atoms or the presence of aromatic rings?". We aim to address a more fundamental and general question : Does the hidden representation encode information about the graph-theoretic properties ? We propose a simple, model-agnostic, probing classifier pipeline to interpret GNN embeddings by probing for encoded graph properties (see fig. 1) across various architectures and datasets. We investigate both local properties like betweennes centrality, as well as global properties like average path length. To our knowledge, this is the first work to explore this direction.

We demonstrate the ability of diagnostic classifiers to effectively highlight graph-theoretic properties in GNN learned latent representations (fig. 7). We explore how different regularization techniques (none, $L_2$ weight decay, dropout) affect the representation of graph properties (fig. 14). We compare how various GNN architectures (GCN, R-GCN, GIN, GAT) differ in their ability to represent graph properties, analyzing whether these differences align with their mathematical frameworks (table 6). The approach is validated through applications to toxicity and fMRI datasets, confirming the alignment of probed properties with domain knowledge (table 8), before exploring the pipeline's inferential power, uncovering structural properties that might not yet have been extensively studied (table 23).

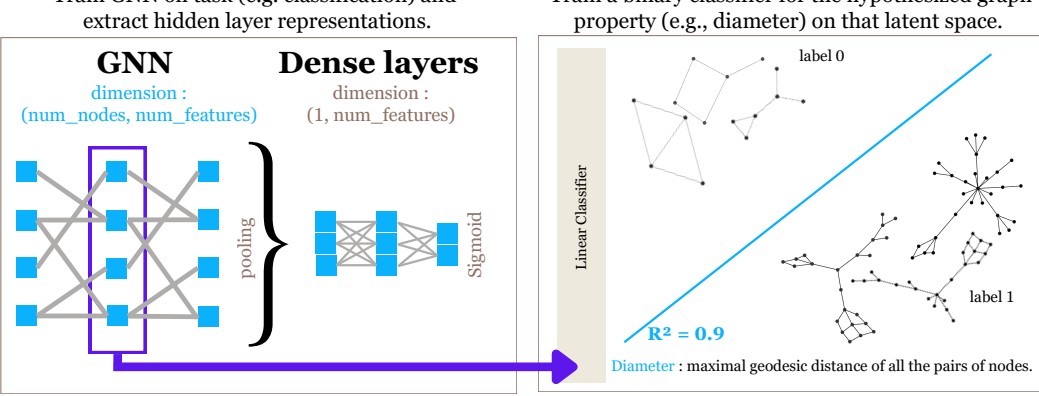

Figure 1: Illustration of the probing pipeline. First, a GNN is trained on a specific task, such as classifying between smaller graphs (label 0) and larger graphs (label 1). Similar to how a CNN might organize images of similar shapes or textures into distinct regions in its feature space, the GNN embeddings might arrange graphs based on structural properties like their diameter. Next, we extract embeddings from the internal layers of the GNN. These embeddings are used to train the probing model—in this case, a binary classifier tasked with determining whether the embeddings encode predictive information about the graph's diameter. If a linear probe has good performance ($R^2$ score) then there exists a hyperplane in the representation space that separates the inputs based on the property

## 2 BACKGROUND

### 2.1 GRAPH NEURAL NETWORKS

Nowadays we have some theoretical understanding of the representational restrictions and capabilities of Graph Neural Networks (GNNs) with regard to the Weisfeiler-Lehman test (Akhondzadeh et al., 2023). We know that these cannot capture certain graph properties, such as connectivity or triangle-freeness (Franks et al., 2024; Kiefer, 2020; Kriege et al., 2018), due to their reliance on local structure. This constraint is also present in (message passing) GNNs.

**Graph Convolutional Network** (GCN) (Kipf & Welling, 2017) are GNNs where for a single layer, the node representation is computed as: $\boldsymbol{X}' = \sigma\left(\tilde{\boldsymbol{D}}^{-1/2} \cdot \tilde{\boldsymbol{A}} \cdot \tilde{\boldsymbol{D}}^{-1/2} \cdot \boldsymbol{X} \cdot \boldsymbol{W}\right)$. We know that GNNs which rely solely on local information, like the **GCN** and its relational variant (**R-GCN**) (Schlichtkrull et al., 2018), cannot compute important graph properties, such as girth and diameter or eigenvector centrality (Garg et al., 2020). We are therefore also investigating more globally aware networks like **GAT** (Graph Attention Network) (Veličković et al., 2018) and **GIN** (Graph Isomorphism Network) (Xu et al., 2019).

GAT makes use of self-attention and is thereby more expressive than the GCN. However, its reliance on feature-dependent weights and structure-free normalisation limits its ability to capture specific structural properties that do not directly depend on edges. This is particularly true for tasks where node features alone are not enough, and global graph structures are crucial (e.g., tasks requiring knowledge of subgraphs or non-local patterns). GIN aggregates node features in a way that mimics the Weisfeiler-Lehman test. By using the MLP equivalent to an Injective Update Function, GIN avoids oversimplifying the aggregation step, making it as expressive as the WL test. Thus, it is likely to excel at encoding complex graph properties and solving classification tasks.

### 2.2 GRAPH PROPERTIES

Graph theory is a branch of mathematics that studies the properties and relationships of graphs. Graphs can be undirected or directed and analysed through both local and global properties. Local properties (like node degree or clustering coefficient) are based on the neighbors of a node. In contrast, global properties (such as diameter and characteristic path length) assess the overall graph

structure. Global graph properties can be associated with higher level complex systems' characteristics like the presence of repeated motifs in the graphs or information-flow properties. See the appendix B for a list of local and global properties used in our experiments.

We can distinguish different global properties, *basic* ones like the number of nodes a graph has, *clustering and centrality* ones, *graph motifs and substructures*, *spectral and small-world properties*. As an higher-order analysis, the recurrence of specific motifs within network substructures—such as triangles, cliques, or feed-forward loops can be seen as the fundamental building blocks that dictate the system's functionality and resilience. Small-worldness, as characterised by Barabási (Albert & Barabási, 2002), reveal how networks can maintain short path lengths despite their expansive size and sparse connectivity. This kind of higher order properties are very interesting in order to understand how the macroscopic behaviour of complex systems emerges from the intricate interplay of their microscopic components (Barabási et al., 2002). For example how diseases spread in social networks, how neurons interact in the brain, or how information propagates through the Internet.

GNNs synthesise local topological features into global structures, abstract these representations into higher-order graph attributes. Each layer progressively expands the receptive field, mirroring how hierarchical feature learning works in convolutional neural networks (CNNs) for images. Probing their learnt representations should act as a scalable proxy to investigate how global arrangement and connectivity patterns influence a system's function. In other terms, by dissecting these learned embeddings, we can possibly delve into the intricate relationships between a network's macroscopic arrangement and its emergent behaviours. Based on the message passing paradigm in GNNs, as layers progress, one would expect an increased abstraction in the selection of graph properties. Initially, local features like node degree dominate, but deeper layers progressively capture more global properties, such as connectivity patterns and centrality. Through hierarchical pooling or readout mechanisms, GNNs can aggregate node embeddings into a single, global graph-level embedding. Graphs that share structural similarities or patterns of interaction among nodes are organised closely in the embedding space, allowing the model to differentiate between classes of graphs, such as those with and without long paths (fig. 1). This is why, AI engineers are likely to focus on the validation framework, emphasizing the alignment between probing results and the predictions derived from the message-passing paradigm—particularly insights revealed in the initial layers of the model. In contrast, from the standpoint of domain researchers—such as chemists or neuroscientists— the most compelling aspect is found in the later layers of the model, where the abstract representations become increasingly capable of rendering the problem linearly separable, thus facilitating clear interpretability of classification decisions and offering domain-specific interpretations.

## 2.3 PROBING CLASSIFIERS

In prior work (Hupkes et al., 2018) probing classifiers have been used for linguistic properties. Here, we adapt them for graph features. Unlike unsupervised techniques such as PCA or T-SNE, which are useful to visualize input data with regard to the embedding latent space, we adopt a supervised framework to quantitatively assess how specific properties are encoded within the embedding space of GNNs. Let $g : f_l(x) \mapsto \hat{z}$ represent a probing classifier, used to map the learned intermediate representations from the original model $f$ to a specific property $\hat{z}$. The choice of a linear classifier for $g$ is motivated primarily by its simplicity. If a linear probe performs well, it suggests the existence of a hyperplane in the representation space that separates the inputs based on their properties, indicating linear separability.

Another advantage of a simple linear probe is avoiding the risk that a more complex classifier might infer features that are not actually used by the network itself (Hupkes et al., 2018). While other non-linear probes have been explored in the literature (Belinkov, 2021), even studies showing improved performance with complex probes maintain the same logic: $\mathrm{Perf}(g, f_1, \mathcal{D}_O, \mathcal{D}_P) > \mathrm{Perf}(g, f_2, \mathcal{D}_O, \mathcal{D}_P)$ holds across representations $f_1(x)$ and $f_2(x)$ when evaluated by a consistent probe $g$. This consistency ensures valid comparison, underscoring that if a property can be predicted well by a simple probe, it is likely relevant to the primary classification task. From an information-theoretic perspective, training the probing classifier $g$ can be viewed as estimating the mutual information between the learned representations $f_l(x)$ and the property $z$. This mutual information is denoted as $I(\mathbf{z}; \mathbf{h})$, where $\mathbf{z}$ refers to the property and $\mathbf{h}$ represents the intermediate representations (Belinkov, 2021).

This supervised approach allows us to define hyperplanes or higher-dimensional decision boundaries that partition the embedding space according to the chosen graph property. The $R^2$ score serves as this information-theoretic measure indicating how well the hyperplane divides the inputs in the embedding space. The R2 score (, see appendix A for a formal definition) measures the proportion of variance in the dependent variable that is predictable from the independent variable(s). A $R^2$ near 1 indicates that the embeddings are highly informative about $\hat{z}$, suggesting that the neural model has internalized this property in a linearly accessible manner.

By defining specific properties that could divide the embedding space and assessing how well the corresponding hyperplanes make the embedding space linearly separable, we gain quantitative insights into the abstract features aggregated within the embeddings. This method moves beyond mere hypothesis generation based on clustering patterns observed through techniques like PCA, providing a rigorous framework for understanding how well the embedding space represents complex graph properties. It can also be thought as complementary from the T-SNE and PCA visualisation techniques, as it provides a quantitative measure of the separability of the embeddings based on hypothesised properties of interest. We illustrate the evolution of the separability of graphs in the embeddings in figs. 4, 7 and 16 using a T-SNE visualisation and the corresponding separability with the properties thanks to the probing.

## 3 DATASETS

All three datasets have the same setup: given a set of graphs $\{\mathcal{G}^1, \mathcal{G}^2, \ldots, \mathcal{G}^N\}$, predict the corresponding binary labels $\{y^1, y^2, \ldots, y^N\}$.

**The Grid-House dataset** inspired by (Agarwal et al., 2023) is designed to evaluate the compositionality of GNNs. It features two concepts: a 3x3 grid and a house-shaped graph made of five nodes. The dataset consists of Barabási-Albert (BA) graphs (Barabási, 2009) with a normal distribution of the number of nodes. The negative class includes a BA graph connected to *either* a grid or a house, while the positive class contains a BA graph connected to *both* a grid and a house (see fig. 2). In order to ensure that the average number of nodes is the same between classes, the number of nodes is a uniformly distributed between 6 and 21 for the grid graphs, between 7 and 22 for the house graphs, and between 1 and 16 when both are present. During generation, we ensure no test set leakage by removing isomorphisms. On 2,000 graphs, we perform an 80/20 train/test split.

For accurate classification, models need to identify and combine simple patterns. Recognizing isolated patterns or single node features is not sufficient. The dataset helps investigate how GNNs combine multiple concepts and addresses the "laziness" phenomenon, where networks learn patterns characterising only one class and predict the other by default (Longa et al., 2023b).

The dataset has been structured such that an optimal, linearly separable solution requires the combination of local properties, such as eigenvector centrality and betweenness centrality, or the identification of global structural motifs, like counting the number of squares (i.e., four-node cycles). A random Barabási-Albert graph can't contain any four-node cycles, while a grid subgraph will consistently exhibit four such cycles. A house subgraph contains exactly one four-nodes cycle and one three-nodes cycle. Therefore, a graph that contains both a grid and a house will have a total of five four-node cycles. The presence of a three-node cycle could help the diagnostic of one type of graph in the negative class but is not necessary nor sufficient for solving the classification problem. On the contrary, counting the number of four-node cycle is necessary and sufficient. Thus, distinguishing between the classes does not really necessitates leveraging centrality-based measures but only recognizing the presence of a specific number of four-node cycles, enabling the model to effectively differentiate between the positive and negative classes. Thus the interesting results of fig. 4.

The Grid-House dataset serves a critical purpose in our study: it offers a controlled and well-defined environment for rigorously testing our hypotheses. The simplicity of these controlled constraints allow us to verify whether the GNN operates as intended in a setting where extraneous factors are minimized. If a model struggles with this dataset, it would likely underperform on real-world, more complex graphs, underscoring its diagnostic value.

**ClinTox Molecular** contains molecular graphs representing compounds with binary labels indicating whether they are toxic or non-toxic. The dataset consists of 1,491 drug compounds with

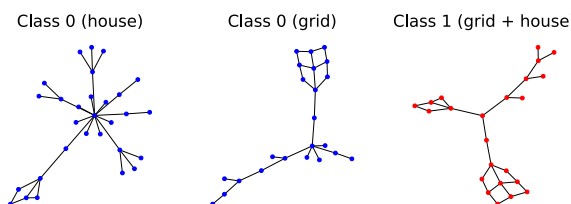

Figure 2: Illustration of the grid-house dataset. The first class (0) include graphs with either a house (square+triangle) either a 3x3 grid (4 squares). The second class include both a house and a grid.

known chemical structures. Each molecule is represented as a graph where nodes correspond to atoms and edges to bonds, with node features representing atom types and edge features representing bond types. The task is to predict toxicity.

**fMRI FC connectomes** consists of two parts. The *Autism Brain Imaging Data Exchange I* dataset contains 528 ASD patients and 571 typically developed (TD) individuals, the *REST-meta-MDD* dataset contains 848 MDD patients and 794 healthy controls. For both, the task is to classify these. We use the datasets with functional connectivity (FC) graphs, as prepared by Zheng et al. (2023). We perform a 95/5% train-test random split.

## 4 METHODOLOGY

For each of the three datasets, we use a similar network architecture consisting of a number of GNN (GCN, GIN, or GAT) layers, followed by a pooling operation (mean- (Kipf & Welling, 2017), sum- (Xu et al., 2019), or max-pooling (Hamilton et al., 2017)), and then a number of dense layers. We optimized the hyperparameters to obtain the best models possible. For the **Grid-House dataset** the complete hyperparameter information can be found in table 4. We ran each model 20 times and took the one with the best accuracy. As expected, there is a correlation of 0.992 between task accuracy and the maximum probing accuracy as seen in fig. 6.

We compared different regularization methods. The explicit $L_2$ **regularization** encourages the network to keep the weights small, and we expect that this would make the embeddings less sensitive to fluctuations in the input data which would translate in later layers being more selective to graph properties. **Dropout** randomly disables a fraction of the neurons during each training iteration which forces the network to learn redundant representations, as any neuron could be dropped out. These redundant representations might make it more difficult to linearly separate the graph embeddings. We expect later layers to distinguish less between graph properties. We plot only post-pooling layers for the sake of clarity.

For the **ClinTox Molecular** dataset, we ranged the number of layers from 4 to 6 and hidden dimensions from 64 to 256. The final model architectures were selected based on optimal performance on the ClinTox dataset. For **fMRI FC connectomes** the hyperparamter search space is described in table 15.

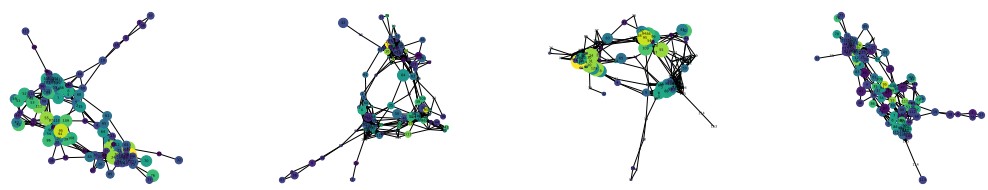

Figure 3: Visualization of test set graph in ASD dataset with node sizes based on (from left to right): (a) degree, (b) betweenness centrality, (c) eigenvector centrality, and (d) clustering coefficient.

**Probing** is performed on the train and test sets, where train features $\{f_{\text{train}}^{(i)}\}$ and graph properties $\{z_{\text{train}}^{(i)}\}$ are paired for each graph (equally for the test set). Let's define at least one example for the *GCN* model. Let $\mathcal{G}^i = (A^i, X^i)$ denote the $i$-th graph, where $A^i$ is the adjacency matrix and $X^i$ is the node feature matrix as previously defined. The GCN layers iteratively update the node features $H^{(l)}$ through graph convolutions defined previously as $H^{(l+1)} = \sigma(\hat{A} H^{(l)} W^{(l)})$, where $\hat{A}$ is the normalized adjacency matrix, $W^{(l)}$ are the trainable weights, and $\sigma$ is a non-linear activation function (ReLU). The node embeddings $H^{(l)}$ at each layer $l$ capture both local and global structural information by aggregating features from neighboring nodes. The final node embeddings $H^{(4)}$ are pooled using global max pooling to generate a graph-level embedding $H_{\text{global}}$, which is passed through three fully connected layers to produce the final prediction $\hat{y}$. We define these post pooling operations as $H_{\text{global}}^{(5)} = \sigma(W_1 H_{\text{global}})$, $H_{\text{global}}^{(6)} = \sigma(W_2 z_1)$, $\hat{y} = \text{softmax}(W_3 z_2)$. For probing purposes, we use $H^{(l)}$ at different layers to evaluate node-level properties, while $H_{\text{global}}, H_{\text{global}}^{(5)}, H_{\text{global}}^{(6)}$, and $\hat{y}$ are used to assess graph-level properties.

We aggregate node embeddings across all graphs to train a single probing classifier for each graph property. For each property, we construct a feature matrix by combining embeddings across all graphs, layer per layer. The classifier $g$ is then trained on this aggregated dataset to predict graph properties $z_k^{(i)}$, where $i$ denotes the $i$-th graph and $k$ represents the $k$-th graph property, as defined in table 3. This approach assumes that the relationships between node or graph embeddings and properties are consistent across graphs.

Probing pre-pooling layers to predict global graph properties presents challenges due to the varying numbers of nodes across graphs and the individual states for each node. To handle this, one approach would involve concatenating and flattening the embeddings into a matrix with dimensions (number of nodes, number of features), padding with zeros if a graph has fewer nodes than the maximum in the dataset. However, flattening introduces issues because nodes do not have a canonical ordering; instead, they follow an arbitrary order based on their appearance in the dataset. This inconsistency can undermine permutation invariance, especially since a simple linear classifier applied to the flattened embeddings is not inherently permutation invariant.

To address this, we compare two methods. One where we mean-pool the node embeddings before probing them like in fig. 12. But we might miss relevant basic properties. At the same time, we make sure that our probe is not extrapolating more structural information that what is encoded in juxtaposed node embeddings. A probing classifier, by operating on the set of embeddings simultaneously, could learn to infer relationships and patterns between nodes, leveraging the contextual relationships implicit in their embeddings. The other one where we first sort the embeddings in descending order based on their norms before concatenating fig. 4. This ordering depends only on the inherent properties of the embeddings themselves, not on their original ordering in the graph. As such, it inherently respects permutation invariance because reordering the nodes does not affect their norms or the resulting sorted order. Conveniently, sorting in this way ensures that any padding zeros align at the end of the sequence, enabling learnable representations for graphs with varying node counts. While sorting for permutation invariance is not widely discussed in the literature, it provides a practical solution by using the embeddings' properties to enforce consistent ordering across graphs.

## 5 RESULTS

### 5.1 GRID-HOUSE DATASET

The models achieved high performance scores, ranked according to their expressivity: GCN scored 0.90, GAT 0.97, and GIN 1.00 (table 5). The probing results demonstrate that the *number of squares* consistently yields the highest $R^2$ scores across all models in the global graph embeddings, both with mean-pooled GNN layer and without. This confirm our initial hypothesis where the number of square is the property of interest to perform this classification, as show the correlation between the highest R2 scores (*# squares*) and the performance of models fig. 6.

When comparing the mean-pooling and norm-sorting methods, the key observation is that mean-pooling renders the probe unable to predict *basic properties* such as the number of nodes. It signifi-

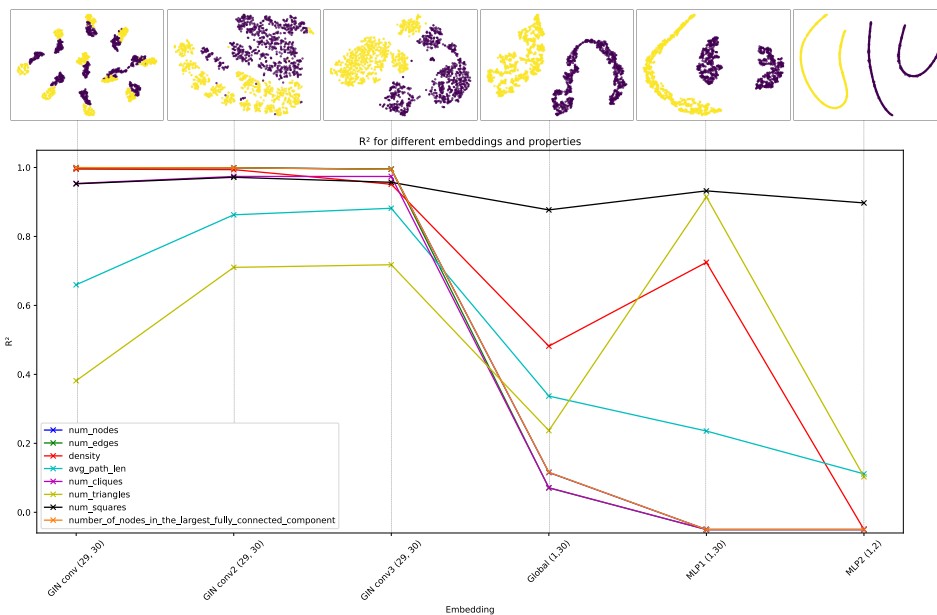

Figure 4: T-SNE visualisation across different layers of our GIN architecture aligned with the probing $R^2$ scores plots (Grid House)

cantly reduces the probe's ability to predict *clustering and centrality*-related properties and slightly diminishes its performance in predicting higher-order properties, such as the *# square*.

In general, the spectral and spatial perspectives are confirmed as deeper layers further smooth the signal, capturing global properties but potentially losing high-frequency details. Analogically, they extend their receptive field, it's well visible with the message passing GCN fig. 8, incorporating more distant nodes into each node's representation as k-hop neighborhood are taken into account. As seen in the T-SNE visualizations figs. 8, 12 and 17, the final layers tend to focus on the *# squares*, effectively partitioning the graphs into two classes: those with #squares $< 5$ (indicating either the grid or house alone) and those with #squares $= 5$ (indicating the presence of both substructures). This reduction in feature space through the layers aligns with the model's goal of optimizing the decision boundary for binary classification, where the number of squares becomes a clear and dominant factor for separability. Interestingly enough, the GIN model has its layer *MLP1* partitioning the graphs *label 0* between those with a house and those with a grid using *# of triangle* significantly. The Gat model present a form of confusion between classes at the layer *Global* before clearly sharping its representation and classification abilities. The other present properties, *density* and *average path length* are also prominent and make sense as the presence of both a house and a grid does slightly increase the average density and path length of graphs. These findings confirm the correspondence between graph embeddings clustering and property hyperplane separation.

We further observe that, for both the GCN and GIN models, the application of $L_2$ regularization yields the expected behavior. The last layer of the GCN in fig. 10 shows a stronger dominance of the number of squares feature when $L_2$ regularization is applied compared to when it is not. Similarly, in the GIN fig. 14, both *# triangles* and *density* become less detectable relative to the number of squares, by the probing classifier in the final layers under $L_2$ regularization, consistent with the anticipated effects on the feature representation. Dropout leads to the representation of multiple properties in the final layers. Notably, in fig. 11, the last layer exhibits a reduced separability gap between the *# square* and other properties, indicating a more distributed feature representation when dropout is applied.

We can use the *number of square* R² score of models to compare their expressivity. For the GCN fig. 8, the slow increase of the R² score correspond to the message-passing theory as we said. The use of the density as a side property make sense but also indicate the lack of optimization of this model,

which we can attest by looking at the T-SNE of the last layer where some graphs are misgrouped. The regularization seems to correct this effect partialy fig. 10.

The GAT and GIN models shows a strong dominance of *of square* from the beginning. Either with an Attention-Weighted Aggregation or a MLP injective aggregation, we don't expect such score from the first layer. The receptive field is limited to 1-hop nodes. This could be explained by a form of signal short-cut where local features like the degree of neighbors is already enough to expect a four-cycle pattern. Similarly, the closeness centrality of a node or its clustering coefficient is barely encoded in the first layers table 7. The effectivity of both the GAT and the GIN models would then catch these short-cuts very effectively.

The GAT model aggregate the neighbors' information in a weighted manner. This feature-dependent mechanism introduces flexibility but also makes GAT's performance contingent on the quality and richness of the node features. It seems that GAT's broader capability compared to the GCN comes at the cost of "focus", as GAT tends to incorporate multiple features, this may dilute its ability to pinpoint the most crucial property (in this case, the number of squares) for the classification task. This over-reliance on feature aggregation can lead to inefficiencies when simpler, more targeted properties suffice, as seen with GIN. The GIN consistently performs the best on squares with a max $R^2 = 0.93$. The GIN in general is sharper in the aggregation of global graph properties has it shows results only for the three properties of interest (#square, #triangle, density) before filtering them out in the last layer. It highlights that GIN excels at global feature detection and effectively isolates and leverages the most relevant structural property for the task, making it sharp in its ability to simplify complex graph data into essential information for decision-making. In other terms, its reliance on minimal yet critical features reflects its capacity for highly targeted feature extraction. This crucial effectiveness justifies prioritizing it from a research domain perspective, aiming to align with existing domain knowledge or uncover novel structural properties that explain a macro-level attribute.

## 5.2 CLINTOX MOLECULAR

As expected, the GIN model outperform the other models with a test accuracy of 0.93 table 8. When looking at the corresponding linear probing performance in table 1, we find that the highest scores are consistently yielded by the *average degree*, the *spectral radius*, the *algebraic connectivity* and the *density*, in that order.

Table 1: Linear Probing $R^2$ Performance Across GIN Layers for Selected Graph Properties (ClinTox Dataset). Best Scores in Bold; Non-convergence indicated by —

| GIN Layer | Avg. degree | Spectral radius | Alg. co. | Density | Avg. btw. cent. | Graph energy |
|---|---|---|---|---|---|---|
| x_global | **0.81** | 0.74 | 0.67 | 0.58 | 0.48 | 0.44 |
| x6 (MLP) | **0.80** | 0.74 | 0.66 | 0.58 | 0.42 | 0.44 |
| x7 (MLP) | **0.75** | 0.71 | 0.56 | 0.50 | 0.47 | 0.46 |
| x8 (MLP) | — | **0.07** | 0.02 | 0.00 | 0.06 | 0.05 |

These findings validate our methodology on already known domain knowledge. Indeed, the average degree of atoms in a molecule provides a straightforward interpretation, as atoms with higher valencies are generally less stable and less biologically compatible. For instance, hydrogen with a valency of 1 and oxygen with a valency of 2 are more compatible with carbon-based molecules, whereas sulfur, with a valency of 6, is less favorable for biological systems (Komarnisky et al., 2003). Therefore, the average degree serves as a useful indicator of molecular toxicity. Additionally, the spectral radius, often associated with molecular stability and reactivity, is another valuable graph property. Molecules with a lower spectral radius tend to be more stable, while those with a higher spectral radius may exhibit localized electron densities, increasing their reactivity. Using this property to predict molecular toxicity is a logical approach.

## 5.3 FMRI FC CONNECTOMES

Here, our primary focus is to uncover properties under exploited by the field. GIN outperforms other architectures table 14, reproducing the observation by Zheng et al. (2023). It makes sense to study

exclusively the last layers of the GIN model. The probing results on the **ASD** dataset reveal that the *number of triangles* consistently achieves high $R^2$ scores across all models, with particularly strong performance in GIN models. This property is followed by the *spectral radius* and the *density*.

For the **MDD** results, detailed results tables 20 and 22 to 24 reveal that the *number of triangles* still consistently achieves high $R^2$ scores across all models while being less of a distinctive feature than in ASD. This time, the *spectral radius* is dominated by the *density* of the graph. In general, the embeddings from the 7th layer of our GIN architecture exhibit higher $R^2$ scores for relevant graph properties, suggesting improved separability in the embedding space for MDD classification compared to ASD. This indicates that the learned representations at this depth capture more discriminative structural features, facilitating more effective class separation between MDD and healthy controls.

## 6 DISCUSSION

### 6.1 EXPECTATIONS

With **Grid-House** we hypothesized that the GNNs would benefit from leveraging the *of square* to render the problem linearly separable. Based on their mathematical restrictions, we hypothesized that the GIN would perform better than the GAT and the GCN. Regularization methods should either refine the representations, as seen with $L_2$ regularization, or distribute them more broadly, as achieved with dropout.

Based on the message-passing paradigm, we anticipated a clear absence of the *of square* in the first layer. Additionally, we expected that the mean-pooling and norm-sorting methods would not significantly alter how representations are probed, except for basic properties like the number of nodes, which are easily interpretable from the tensor of node vectors but not from an aggregated representation.

For the **ClinTox Molecular** dataset, based on the literature (Kengkanna & Ohue, 2024; Chen et al., 2021; Jiang et al., 2021) some few properties have been found to be link with toxicity such as the node degree (i.e. the valency), subgraph patterns (functional groups, chemical fragments), and the overall graph connectivity.

Based on existing literature on functional connectivity (FC) network properties in ASD and MDD appendix G, we hypothesized that specific properties would be critical in classifying brain networks for the **fMRI FC connectomes** dataset. For ASD, we expect *betweenness centrality* to play a significant role at the node level, reflecting local overconnectivity. At the graph level, we anticipate that *clustering coefficient*, *characteristic path length*, and *small-worldness* will be essential in capturing the local and global network disruptions seen in ASD, particularly the imbalance between local overconnectivity and long-range underconnectivity. For MDD, we hypothesise that increased *clustering coefficients*, *modularity*, *number of triangles* and *number of squares* will be key features for classification, as they could indicate of heightened local interconnectedness and disrupted global integration.

### 6.2 FINDINGS

We first demonstrate the feasibility of our probing method through the **Grid-House** dataset. In line with our expectations, The *number of squares* metric dominated across all layers and models, with GIN showing enhanced expressivity. Secondary properties like *avg_path_lenght* figs. 7 and 8 showed early significance but gradually diminished through the layers, demonstrating how GNNs act as low-pass filters on graph signals, the receptive field select the most discriminative properties. These findings align with the labels, as the density or the average path length is increase when both a grid and a house are added to a graph. Mean-pooling and norm-sorting methods exhibit different behaviors that can be explained by the type of information the classifier access : aggregated graph data or detailed node embeddings. The probe might also infer relationships and patterns that are implicitly encoded in individual node embeddings but become explicit when considered collectively.

The analogy with computer vision lead to the question of supervision signal as we would expect unsupervised model to capture structural information more effectively than supervised ones (Kingma

& Welling, 2022). Unsupervised learning algorithms can discover hidden patterns and structures in data without explicit instructions, potentially uncovering relationships that may not be immediately obvious to humans. Following this logic, this ability to find latent structures could be advantageous in capturing complex structural information in graph data and lead to similar breakthought than discovering new particle physics phenomena using clustering and dimensionality reduction on collider data or identifying subtypes of cancer with clustering and PCA in genomic studies.

Using the **ClinTox Molecular** dataset to assess molecular toxicity, we explored how key graph properties, such as the *average degree* and *spectral radius*, are utilized by our GIN architecture. The average degree, closely linked to atomic valency, reflects a molecule's potential for interactions. The *spectral radius* offers a complementary hypothesis, suggesting that the overall structural stability of a molecule, independent of specific atomic features, may also be a key factor in toxicity prediction.

With the **fMRI FC connectomes** dataset, the results provided new insights that extended beyond our initial hypothesis. The prominence of the *number of triangles* highlighted the importance of local structural motifs. This makes sense in the context of functional connectivity, where local overconnectivity in specific brain regions, such as sensory and association cortices, has been observed in individuals with ASD. The strong role of triangle motifs may reflect the tight, redundant local connections that characterize these regions, supporting the hypothesis that local overconnectivity is a key factor in ASD. The *spectral radius* and *density* and *graph energy* being particularly significant is also logical, as these properties are closely related to the overall connectivity strength and the compactness of connections within subnetworks. The outcomes for both the ASD and the MDD datasets showed promising results that should be discussed more deeply with neuroscientists. The results mainly suggest the importance of graph substructures or spectral and small-world properties over more basic graph properties to explain how graph neural networks predict these neurological disorders in the FC matrices of patients. This is a result which, to the best of our knowledge, has never been investigated before.

### 6.3 FUTURE WORK

Our methodology has several limitations. While we addressed dataset issues such as leakage and isomorphic graphs, a key challenge remains the lack of guarantees that GNNs find globally optimal solutions, despite their theoretical capacity as universal function approximators. This is particularly evident in fMRI data, where multiple layers of complexity—from MRI limitations and BOLD signal characteristics to Pearson correlation for functional connectivity—introduce noise and inaccuracies. Investigating additional graph properties like girth or complex motifs could be beneficial. Preliminary work on alternative architectures (e.g., GATv2, GraphSAGE, ChebNet, Set2Set, HO-Conv, DiffPool) has begun but is not yet complete. An extensive exploration of 1-WL, 2-WL and 3-WL GNN equivalent could bolster the paper's contributions by showing clear restrictions and capabilities of these models. As a future encouraging work, the question of supervision signal would be very insightful in understanding what type of properties are learn by unsupervised models.

### 7 CONCLUSION

We demonstrate the relevance of using probing classifier as a model-agnostic explainability method for graph neural networks. We manifest both the expressivity of different GNN architectures and their ability to solve a graph classification problem through optimal feature extraction. They render it linearly separable in the space of their embeddings through the computation of graph properties. We validate domain knownledge with the Clintox Molecular dataset and investigate the possibility of formulating hypotheses on the emergent dependence of complex systems attributes to basic and more higher level structural properties with the fMRI FC connectomes dataset. We explore how the macroscopic behavior of complex systems emerges from the intricate interplay of their microscopic components. There is a manifest emergence of molecular qualities like toxicity with regard to their structural properties like *node degree* (atom valency) and *spectral radius* (the molecule's stability). To explain a macro attribute, there are instances where structural properties may offer more insight than the mere aggregation of element properties. Both can be interconnected, yet the challenge remains in disentangling these relationships.

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

## A    R$^2$ SCORE

We are using $R^2$ as the main metrics. The $R^2$ score (coefficient of determination) measures the proportion of variance in the dependent variable that is predictable from the independent variable(s). For a probing classifier, it would indicate how well the probe's predictions match the actual properties being probed. More formally, $R^2$ is defined as:

$$R^2 = 1 - \frac{\sum_i \left( z^{(i)} - \hat{z}^{(i)} \right)^2}{\sum_i \left( z^{(i)} - \bar{z} \right)^2}$$

Where: $z^{(i)}$ is the ground truth value of the property $\hat{z}$ for the $i$-th data point in the probing dataset $\mathcal{D}_P$. $\hat{z}^{(i)}$ is the predicted value of $\hat{z}$ produced by the probing classifier $g$. $\bar{z}$ is the mean of the ground truth values $z^{(i)}$ over the dataset. The numerator represents the residual sum of squares (how far off the predictions are), and the denominator represents the total sum of squares (the variance in the ground truth values).

An $R^2$ value ranges from 0 to 1, where: $R^2 = 1$ means the probing classifier perfectly explains the variance in the target property (i.e., the learned representations fully capture the property). $R^2 = 0$ means the probing classifier does no better than predicting the mean $\bar{z}$, implying the representations do not capture any useful information about the property. Good R2 score should indicate how the model achieves its behavior on the original task Hupkes et al. (2018).

A good $R^2$ score gives a sense of how well the features at each layer can be separated linearly to predict the target labels. The second reason is that a more complex probe "bears the risk that the classifier infers features that are not actually used by the network" (Hupkes et al., 2018). Of course, other non linear probes have been explored in the literature (Belinkov, 2021). If a few studies observed better performance with more complex probes, the logic remained the same Perf $(g, f_1, \mathcal{D}_O, mathcalD_P) > $ Perf $(g, f_2, \mathcal{D}_O, \mathcal{D}_P)$, of two representations $f_1(x)$ and $f_2(x)$, holds across different probes $g$. The important criteria is to compare the results obtained by the same measurement system. In general, if we can predict one property on one embedding for a given classification problem, then it means this properly is useful for the problem resolution.

From an information-theoretic perspective, training the probing classifier $g$ can be viewed as estimating the mutual information between the learned representations $f_l(x)$ and the property $z$. This mutual information is denoted as $I(\mathbf{z}; \mathbf{h})$, where $\mathbf{z}$ refers to the property and $\mathbf{h}$ represents the intermediate representations (Belinkov, 2021).

## B    LOCAL AND GLOBAL GRAPH PROPERTIES

| | Property | Visual Pattern & Definition | Computational Criteria |
|---|---|---|---|
| **Local** | Degree | How many links a node has which is the simplest form of centrality | Count edges per node |
| | Local clustering Coefficient | Are the neighbours of a node also connected together ? | Count triangles of neighbours / total possible triangles of neighbours |
| | Betweenness Centrality | How much of a bridge between clusters is a node. Removing that node would break many shortest paths. Importance in information flow | Number of shortest paths through node |
| | Closeness Centrality | Being in the middle of the network, the barycenter of the graph. | The average length of the geodesic distances to all the other nodes (inverse sum of shortest paths) |
| | Eigenvector Centrality | Being connected to well connected nodes without necessarily having a large number of neighbours itself; influence based on connections | Recursive definition based on neighbours |
| | PageRank | Nodes with important connections; web-inspired importance | Similar to Eigenvector but with random walk and teleportation |

Table 2: Local Network Properties with definition and computational criteria

| | Property | Visual Pattern & Definition | Computational Criteria |
|---|---|---|---|
| **Global** | Number of Nodes | Graph size; total nodes in the network | Count vertices |
| | Number of Edges | Graph density; total connections in the network | Count connections |
| | Density | Overall graph connectivity; how densely connected | Ratio of actual to possible edges |
| | Average Path Length | On average, how close are nodes to each other? Typical distance between node pairs | Average number of steps along the shortest paths for all possible pairs of nodes |
| | Diameter | Graph span; longest of all shortest paths | Maximum shortest path |
| | Radius | Graph core; minimum distance from central to farthest node | Minimum eccentricity |
| | Transitivity | Triangle density; probability of connected node triplets | Ratio of triangles to triads |
| | Assortativity | Node degree correlations; tendency of similar nodes to connect | Pearson correlation of degrees |
| | Number of Cliques | Dense subgraphs; count of maximal fully connected subgraphs | Number of maximal complete subgraphs |
| | Number of Triangles | Local density; fully connected 3-node subgraphs | Count 3-node cliques |
| | Number of Squares | 4-node patterns; cycles in the graph | Count 4-node cycles |
| | Largest Component Size | Main connected structure; size of biggest connected part | Largest set of connected nodes |
| | Average Degree | Overall connectivity; average connections per node | Mean of all node degrees |
| | Spectral Radius | Dominant graph structure; overall connectivity measure | Largest eigenvalue of adjacency matrix |
| | Algebraic Connectivity | Graph cohesion; measure of how well-connected the graph is | Second smallest eigenvalue of Laplacian |
| | Graph Energy | The eigenvalues capture deviations from regularity in the network. Complete graphs or highly connected networks tend to have higher energies due to the larger magnitude of their eigenvalues. In social networks, biology, and communication networks, graph energy can help assess robustness, synchronizability. | Sum of absolute Laplacian eigenvalues |
| | Small World Coefficient | Balance of clustering and paths; small-world characteristics | Comparison to random graph |
| | Small World Index | Refined small-world measure; comparison to random and lattice graphs | Comparison to random and lattice graphs |
| | Betweenness Centralization | Central node dominance; degree of central bridging node | Variation in betweenness centrality across nodes |
| | PageRank Centralization | Influence concentration; degree of dominant influential nodes | Variation in PageRank values across nodes |

Table 3: Global Network Properties with definition and computational criteria

We are using the Small-World Index, $SWI = \left(\frac{L - L_l}{L_r - L_l}\right) \times \left(\frac{C - C_r}{C_l - C_r}\right)$ in our experiment because it provides a more balanced and robust measure of small-world properties. Unlike the Small-World Quotient: $Q = \frac{C/C_r}{L/L_r}$, which can be sensitive to network size and degree, $SWI$ normalizes both the clustering coefficient and average path length with respect to both random and lattice reference graphs. This dual normalization approach ensures that $SWI$ is less prone to false positives or negatives, making it a more reliable metric for our analysis (Neal, 2017).

## C    LITERATURE REVIEW ON RELATED WORK

Existing post-hoc GNN explanations methods can be classified into two main categories: *instance-level* and *model-level* methods (Barredo Arrieta et al., 2020). See (Agarwal et al., 2023; Dai et al., 2022) for nice reviews on the subject. In the realm of instance based methods, *gradient-based* methods use the gradients of the output with respect to the input or intermediate features to measure the importance of each component of the graph. *Decomposition-based* methods try to decompose the input graph into smaller subgraphs or paths that can account for the output. *Surrogate-based* methods use a simpler, more interpretable model to approximate the behavior of the original GNN and provide explanations based on the surrogate model. And finally *Perturbation-based* methods which perturb the input graph by removing or adding nodes, edges, or features, and observe the changes in the output to identify the influential components. The most mainstream technique, GNNExplainer (Ying et al., 2019) achieves explanation by removing redundant edges from an input graph instance, maximizing the mutual information between the distribution of subgraphs and the GNN's prediction. It is able to provide an explanation both in terms of a subgraph of the input instance to explain, and a feature mask indicating the subset of input node features which is most responsible for the GNN's prediction.

For *model-based* techniques, few methods come to mind (Saha et al., 2022; Azzolin et al., 2023; Vu & Thai, 2020; Wang et al., 2023; Xuanyuan et al., 2023; Yuan et al., 2020; Zhang et al., 2021). The most mainstream method seems to be XGNN (Yuan et al., 2020). The authors of XGNN investigate the possible input characteristics used by a GNN for graph classification. But they formulate the problem as a reinforcement learning problem and generate graph patterns iteratively. Such an iterative approach is often intractable for large graphs. Moreover, it does not allow for both node classification and graph classification explanations, nor does it allow for an investigation of the learning process through the different layers of the GNN. In general, none of the techniques allow for an interpretation of the hidden representations states with graph properties.

# D GRID HOUSE ARTIFICIAL DATASET

## D.1 GRID HOUSE FIGURES

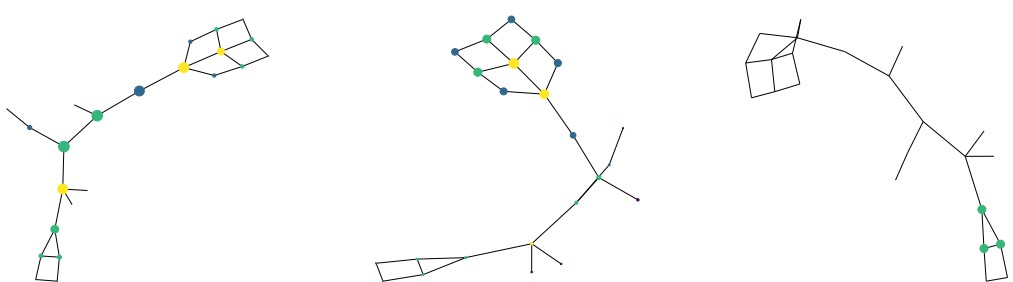

Figure 5: Comparison of different centrality measures for the first graph in our Grid House dataset: (a) betweenness centrality, (b) eigenvector (PageRank) centrality, and (c) local clustering coefficients.

## D.2 GRID HOUSE MODELS

Table 4: Range of Hyper-parameters and Final Specification for the Grid-House Dataset

| Hyper-parameter | Range Examined | Final Specification |
|---|---|---|
| Graph Encoder | | |
| #GNN Layers | $\{[2, 3, 4, 5]\}$ | 4 (GCN), 2 (GIN), 3 (GAT) |
| #MLP Layers | $\{[2, 3, 4]\}$ | 3 (GCN), 2 (GIN), 2 (GAT) |
| Hidden Dimensions | $\{[10, 15, 30, 45, 60, 64, 128, 256]\}$ | 60 (GCN), 30 (GIN), 128 (GAT) |
| Attention Heads (GAT) | $\{[4, 8, 16]\}$ | 8 heads, 32 dimensions per head |
| Learning Rate | $\{[1e-2, 1e-3, 1e-4]\}$ | $1e-3$ |
| Batch Size | $\{[32, 64, 128, 256]\}$ | 64 |
| Weight Decay (when added) | $\{[1e-4, 1e-2]\}$ | $1e-4$ (GCN), $1e-2$ (GIN) |
| Batch Normalization | $\{with, without\}$ | $without$ |
| Dropout (when added) | $\{[0.15, 0.5]\}$ | 0.2 |
| Pooling Method | $\{mean, sum, max\}$ | $max$ (GCN), $mean$ (GIN), $max$ (GAT) |

Table 5: Performance of Different Models with Regularization on the Artificial Dataset (80%-20% Random Split). The highest performance is highlighted with boldface. All performances are reported under their best settings and rounded to 2 decimal places.

| Method | Test Accuracy |
|---|---|
| GCN (control) | 0.90 |
| GCN ($L_2$) | 0.97 |
| GCN (dropout) | 0.93 |
| GIN (control) | **1.00** |
| GIN ($L_2$) | 0.99 |
| GIN (dropout) | 1.00 |
| GAT | 0.97 |

As expected the RGCN outperform the GCN on this node classification task.

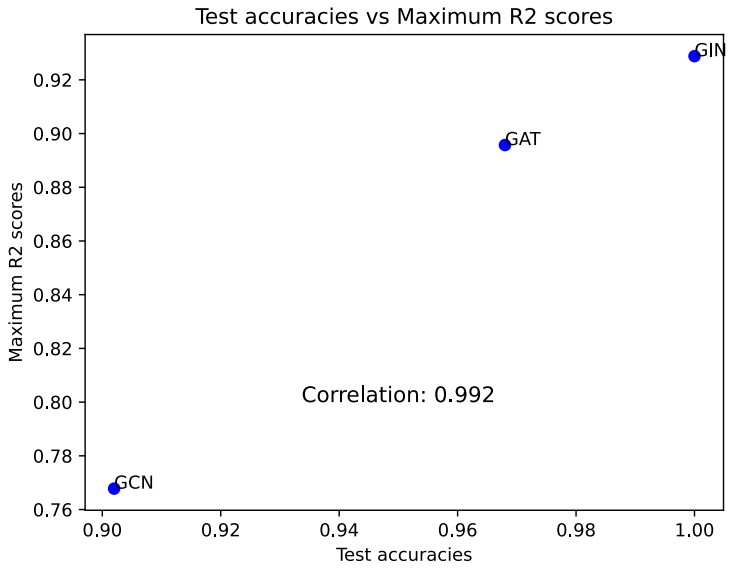

Figure 6: Plot of the correlation between the different model test accuracies and their maximum R2 score (Grid House)

## D.3 GRID HOUSE RESULTS

### D.3.1 GRAPH PROPERTIES PROBING RESULTS

Table 6: Linear Probing $R^2$ Performance Across models for Selected Graph Properties (GridHouse Dataset). Best Scores in Bold; Non-convergence indicated by —

| Model | #nodes | #edges | density | avg path len | #cliques | #triangles | #squares | #Largest Component |
|---|---|---|---|---|---|---|---|---|
| **GCN (control)** | | | | | | | | |
| x_global | 0.36 | — | 0.66 | 0.33 | 0.02 | 0.31 | **0.77** | 0.36 |
| x5 | 0.33 | 0.22 | 0.64 | 0.29 | 0.27 | 0.39 | **0.77** | 0.33 |
| x6 | 0.19 | 0.08 | 0.56 | — | 0.07 | 0.06 | **0.74** | 0.19 |
| x7 | — | — | 0.45 | 0.13 | — | 0.03 | **0.72** | — |
| **GCN ($L_2$)** | | | | | | | | |
| x_global | 0.36 | 0.09 | 0.67 | 0.35 | 0.20 | 0.68 | **0.86** | 0.36 |
| x5 | 0.31 | 0.32 | 0.66 | 0.32 | 0.32 | 0.80 | **0.86** | 0.31 |
| x6 | 0.04 | — | 0.41 | 0.15 | 0.03 | 0.23 | **0.83** | 0.04 |
| x7 | — | — | 0.29 | 0.27 | — | 0.09 | **0.81** | — |
| **GCN (dropout)** | | | | | | | | |
| x_global | 0.21 | 0.07 | 0.67 | 0.33 | 0.07 | 0.63 | **0.72** | 0.22 |
| x5 | — | — | 0.59 | 0.26 | — | 0.66 | **0.74** | — |
| x6 | — | — | 0.42 | 0.21 | — | 0.49 | **0.65** | — |
| x7 | — | — | 0.35 | 0.10 | — | 0.26 | **0.51** | — |
| **GIN (control)** | | | | | | | | |
| x_global | 0.12 | 0.07 | 0.50 | 0.32 | 0.07 | 0.22 | **0.87** | 0.12 |
| x5 | — | — | 0.72 | 0.30 | — | 0.89 | **0.93** | — |
| x6 | — | — | — | 0.02 | — | 0.11 | **0.88** | — |
| **GIN ($L_2$)** | | | | | | | | |
| x_global | — | — | 0.49 | 0.30 | — | 0.18 | **0.85** | — |
| x5 | — | — | 0.51 | 0.15 | — | 0.52 | **0.89** | — |
| x6 | — | — | 0.40 | 0.12 | — | 0.10 | **0.80** | — |
| **GIN (dropout)** | | | | | | | | |
| x_global | — | — | 0.53 | 0.36 | — | 0.25 | **0.87** | — |
| x5 | — | — | 0.71 | 0.33 | — | 0.85 | **0.93** | — |
| x6 | — | — | — | 0.21 | — | 0.34 | **0.91** | — |
| **GAT** | | | | | | | | |
| x_global | 0.54 | 0.59 | — | 0.49 | 0.61 | **0.89** | 0.87 | 0.54 |
| x5 | — | — | 0.33 | 0.27 | — | 0.17 | **0.64** | — |
| x6 | — | — | 0.25 | 0.17 | — | 0.17 | **0.63** | — |

D.3.2 GRAPH PROPERTIES PROBING PLOTS

**GCN**

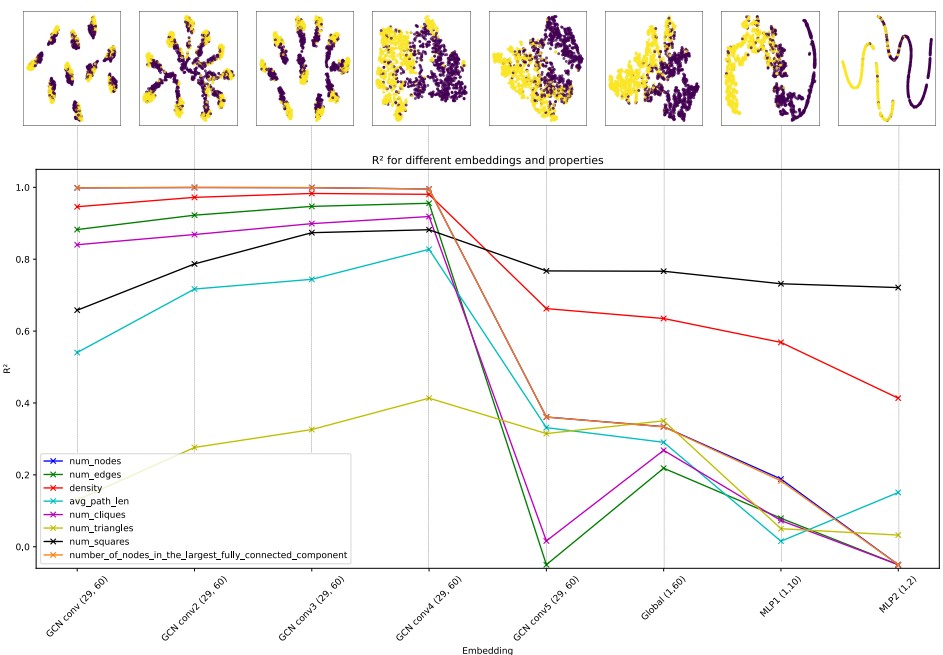

Figure 7: T-SNE visualization across different layers of our GCN architecture aligned with the probing $R^2$ scores plots (Grid House)

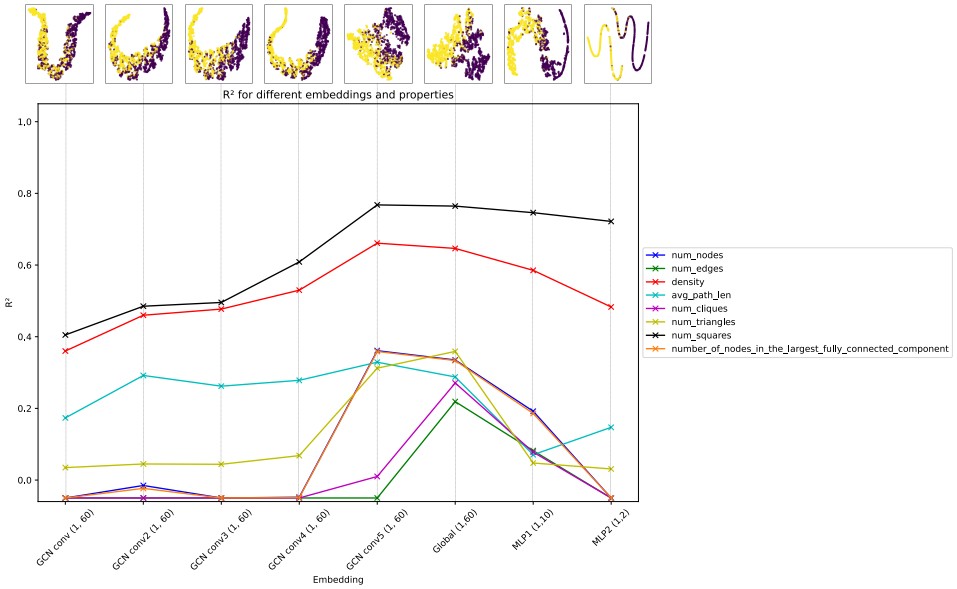

Figure 8: T-SNE visualization across different layers of our GCN architecture aligned with the probing $R^2$ scores plots with mean-pooled node embeddings (Grid House)

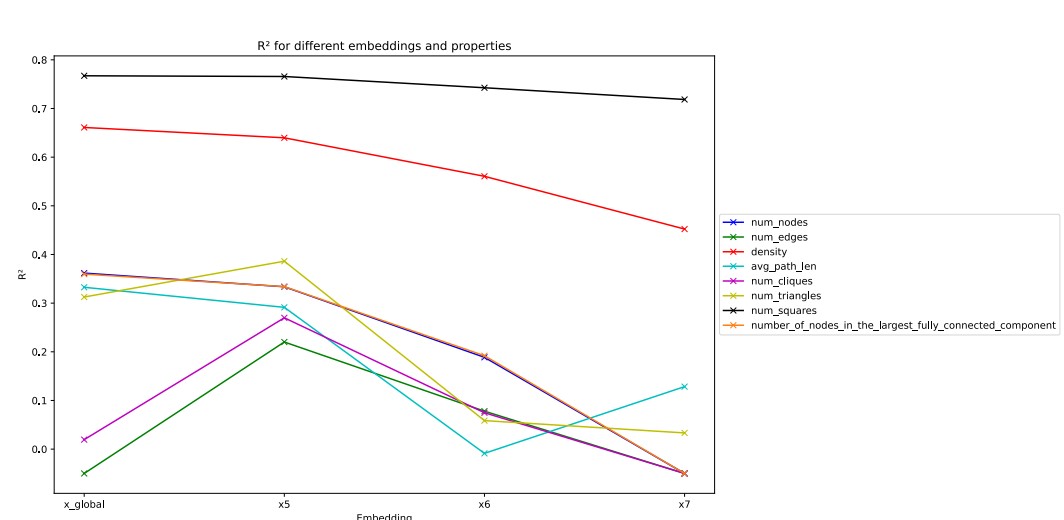

Figure 9: Plot of the GCN (control) $R^2$ results across different layers probing for graph properties with post pooling layers only, allowing clearer visualization and higher order property interpretation (Grid House)

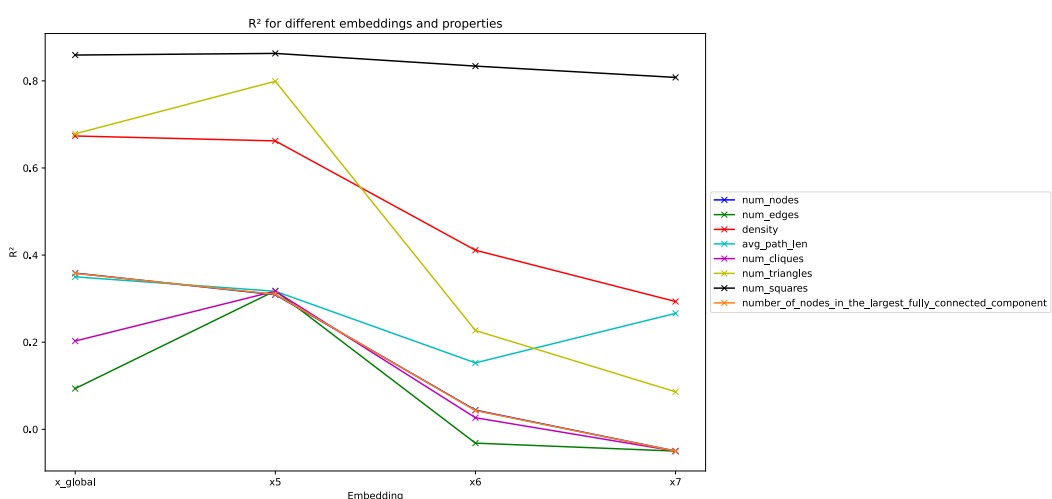

Figure 10: Plot of the GCN ($L_2$) $R^2$ results across different layers probing for graph properties with post pooling layers only, allowing clearer visualization and higher order property interpretation (Grid House)

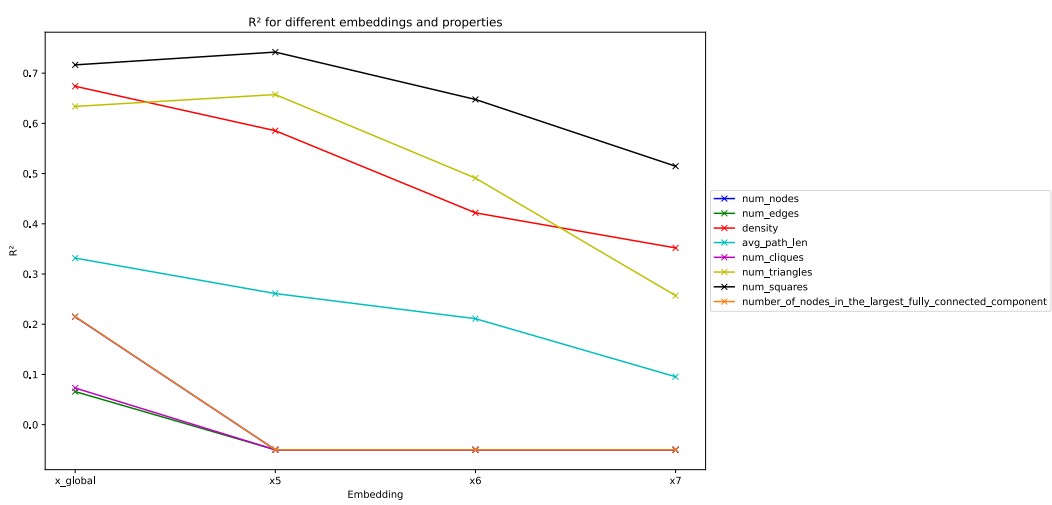

Figure 11: Plot of the GCN (dropout) $R^2$ results across different layers probing for graph properties with post pooling layers only, allowing clearer visualization and higher order property interpretation (Grid House)

**GIN**

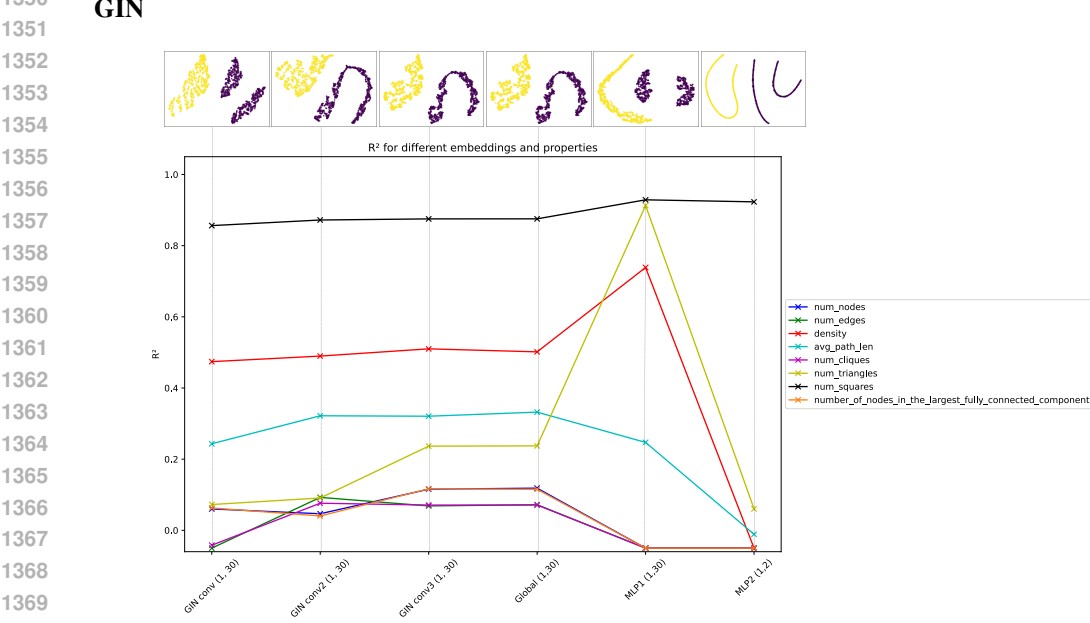

Figure 12: T-SNE visualization across different layers of our GIN architecture aligned with the probing $R^2$ scores plots with mean-pooled node embeddings (Grid House)

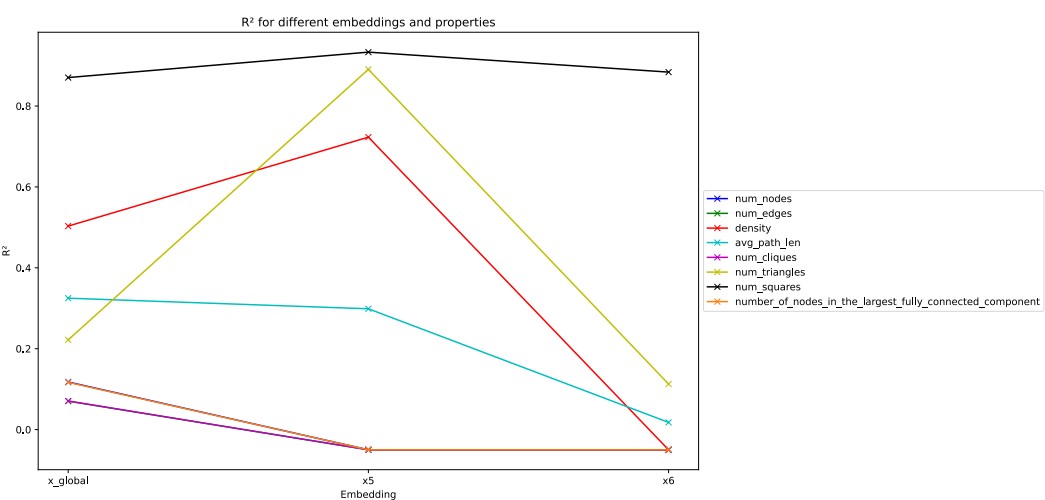

Figure 13: Plot of the GIN (control) $R^2$ results across different layers probing for graph properties with post pooling layers only, allowing clearer visualization and higher order property interpretation (Grid House)

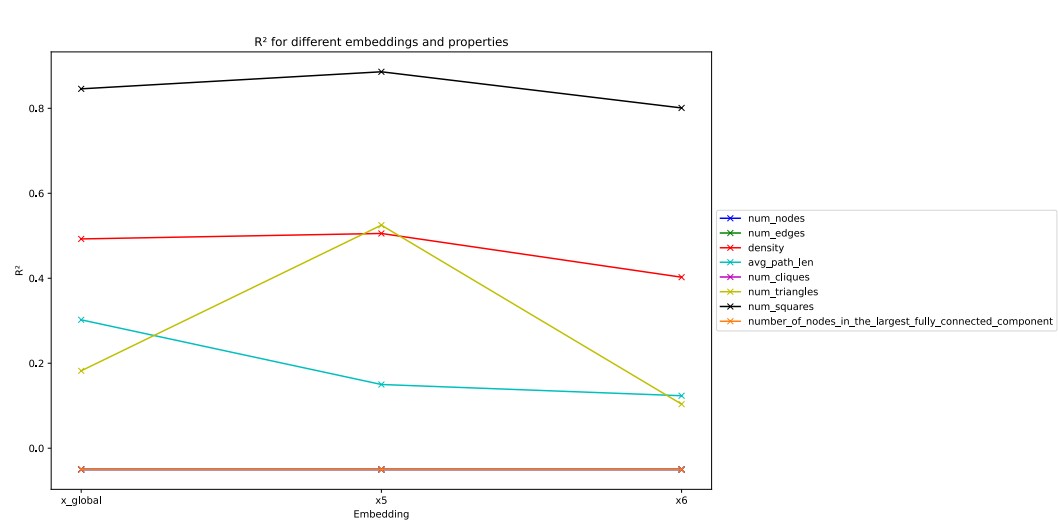

Figure 14: Plot of the GIN ($L_2$) $R^2$ results across different layers probing for graph properties with post pooling layers only, allowing clearer visualization and higher order property interpretation (Grid House)

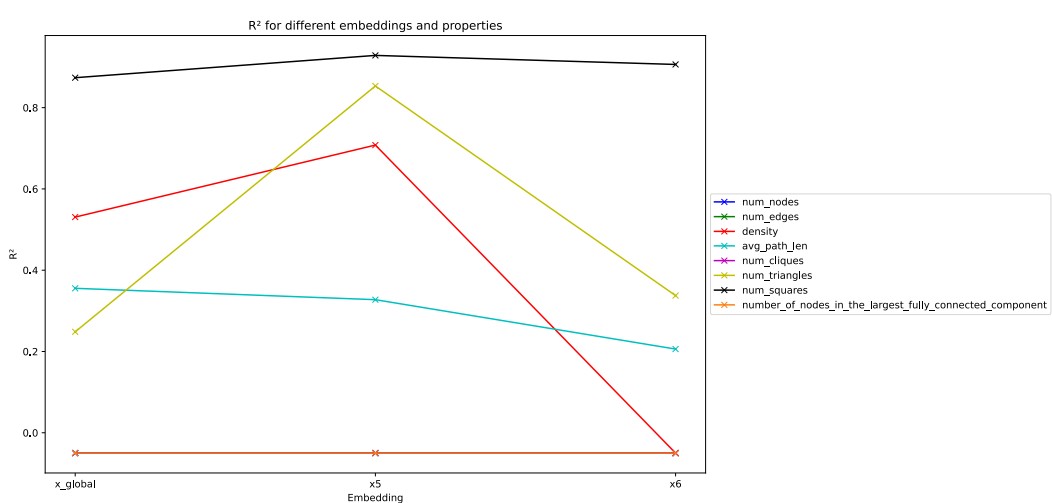

Figure 15: Plot of the GIN (dropout) $R^2$ results across different layers probing for graph properties with post pooling layers only, allowing clearer visualization and higher order property interpretation (Grid House)

**GAT**

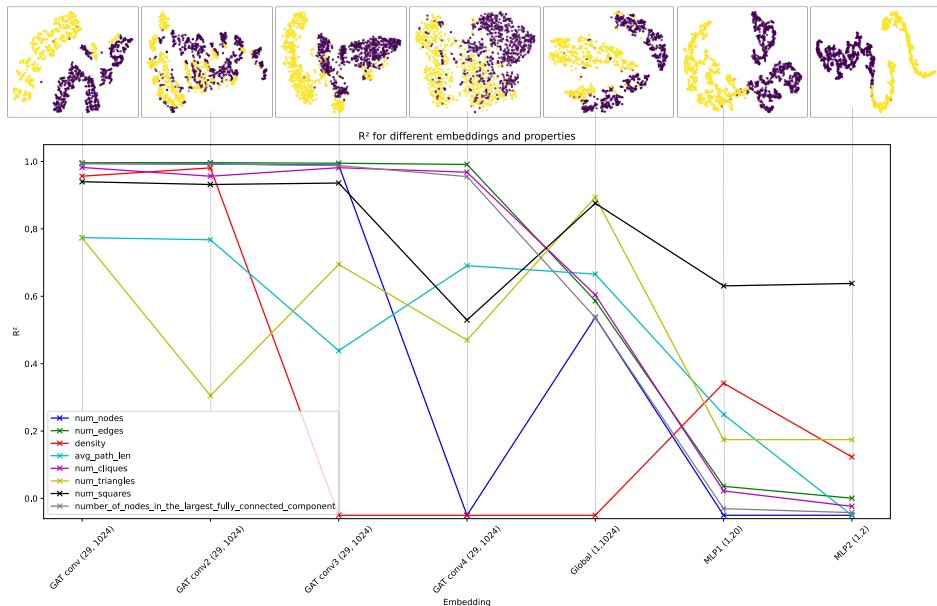

Figure 16: T-SNE visualization across different layers of our GAT architecture aligned with the probing $R^2$ scores plots (Grid House)

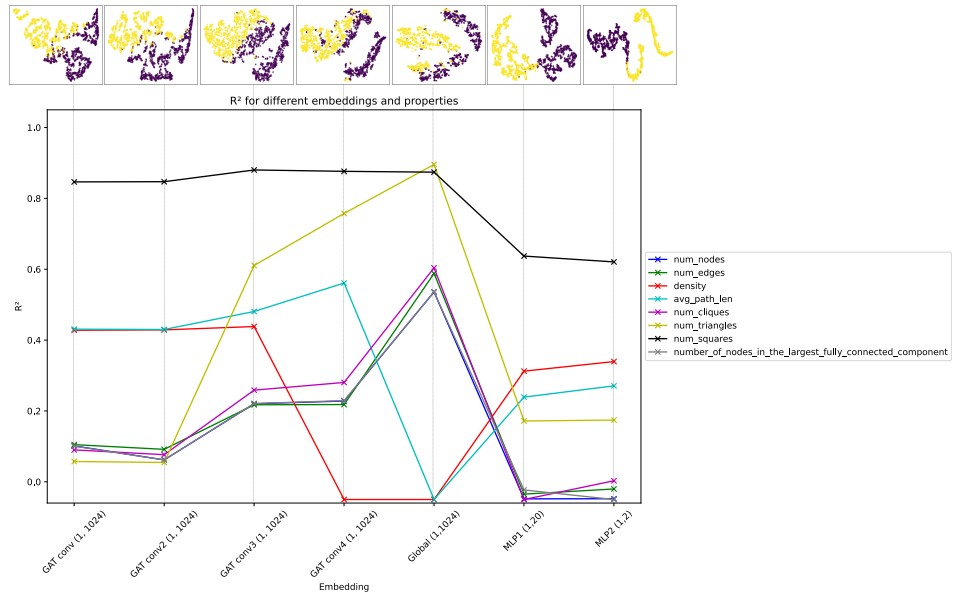

Figure 17: T-SNE visualization across different layers of our GAT architecture aligned with the probing $R^2$ scores plots with mean-pooled node embeddings (Grid House)

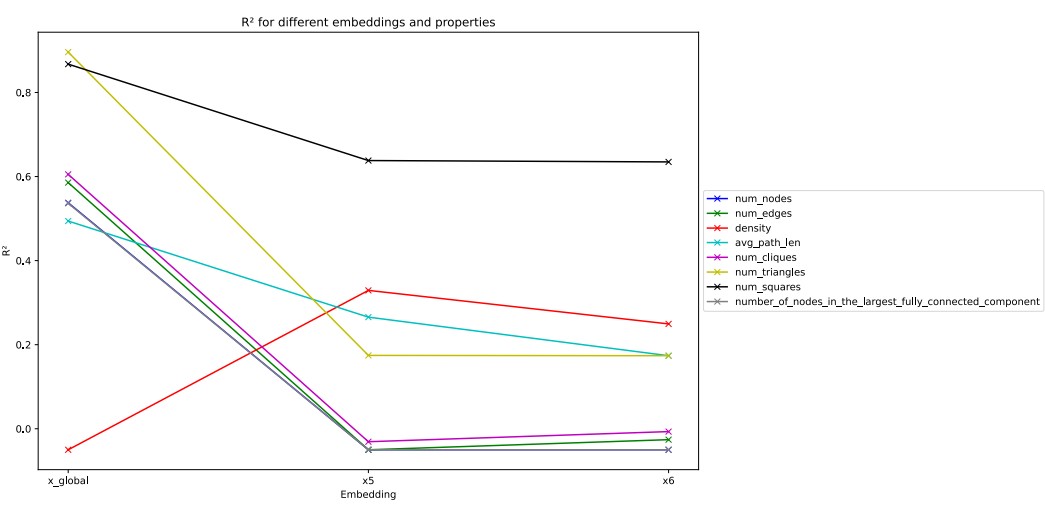

Figure 18: Plot of the GAT $R^2$ results across different layers probing for graph properties with post pooling layers only, allowing clearer visualization and higher order property interpretation (Grid House)

### D.3.3 GRID HOUSE NODE PROPERTIES PROBING RESULTS

Using the probing method developed in the next section, we were not fully able to confirm our initial hypothesis.

Table 7: Linear Probing $R^2$ Performance Across models for Selected Node Properties (GridHouse Dataset). Best Scores in Bold; Non-convergence indicated by —

| **GCN Layer** | degree | closeness | betweenness | eigenvector | clustering | pagerank |
|---|---|---|---|---|---|---|
| x1 (GCN) | 0.50 | 0.22 | 0.25 | 0.19 | 0.06 | **0.56** |
| x2 (GCN) | 0.54 | 0.32 | 0.28 | 0.24 | 0.09 | **0.57** |
| x3 (GCN) | 0.54 | 0.35 | 0.29 | 0.25 | 0.11 | **0.57** |
| x4 (GCN) | 0.55 | 0.37 | 0.28 | 0.30 | 0.17 | **0.57** |
| **GIN Layer** | | | | | | |
| x1 (GIN) | 0.55 | 0.18 | 0.24 | 0.22 | 0.05 | **0.56** |
| x2 (GIN) | 0.52 | 0.34 | 0.27 | 0.25 | 0.07 | **0.54** |
| **GAT Layer** | | | | | | |
| Layer 0 | **0.55** | 0.07 | 0.05 | 0.32 | 0.28 | 0.17 |
| Layer 1 | **0.52** | 0.48 | 0.08 | 0.31 | 0.30 | 0.14 |
| Layer 2 | 0.47 | **0.55** | — | 0.29 | 0.29 | — |
| Layer 3 | **0.41** | — | 0.14 | 0.19 | 0.26 | — |
| Layer 4 | 0.35 | **0.50** | 0.12 | 0.21 | 0.23 | — |

In these pre-pooling layers, we first observe the predominance of *page rank* and *node degree* in the early layers and in all the layers of the GCN and the GIN (which has only two of them). When considering the last layers of the GAT (unfortunately we should have have similar architecture with the GIN in order to fully test our hypothesis) it seems that *closeness*, *node degree* and *clustering coefficient* are the most significant. This aligns with our framing of the graph classification task, which is largely driven by the detection of squares and the fact that pre-pooling layers leading to this property detection should affect mostly these three properties. But this does not align with the use of node properties in a graph in order to do graph classification. This still makes a lot of sense. In general, contrary to the graph probing, and to the exception of the node degree, we see that there is not a single property clearly dominating others but that we go towards a combination of different properties just before the graph pooling method. We would have expect the GIN architecture to show similar results with four layers (as we already see an important increase with regard to the closeness between the first and second layer).

# E    CLINTOX DATASET

## E.1    MODEL

Table 8: Performance of Different Models on ClinTox with a 80%-20% Random Split. The highest performance is highlighted with boldface. All the performance of methods are reported under their best settings.

| Method | ClinTox |
|--------|---------|
| GCN | 0.91 |
| GAT | 0.92 |
| GIN | 0.93 |

## E.2    RESULTS

### E.2.1    GRAPHS PROPERTIES PROBING RESULTS

Table 9: Linear Probing $R^2$ Performance across the GIN layers for basic graph properties (ClinTox dataset). Best Scores in Bold; Non-convergence indicated by —(full)

| GIN Layer | # Nodes | # Edges | Density | Avg. Path Length | Diameter | Radius |
|-----------|---------|---------|---------|------------------|----------|--------|
| x1 (GIN) | **1.00** | **1.00** | 0.66 | 0.76 | 0.55 | 0.60 |
| x2 (GIN) | **1.00** | **1.00** | 0.57 | 0.95 | 0.88** | 0.84 |
| x3 (GIN) | **1.00** | **1.00** | 0.62 | **0.97** | 0.93 | 0.89 |
| x4 (GIN) | **0.99** | **0.99** | 0.37 | 0.91 | 0.82 | 0.82 |
| x5 (GIN) | **0.99** | **0.99** | 0.29 | 0.90 | 0.82 | 0.82 |
| x_global | 0.41 | 0.44 | 0.58 | 0.20 | 0.20 | 0.20 |
| x6 (MLP) | 0.40 | 0.44 | 0.58 | 0.19 | 0.19 | 0.19 |
| x7 (MLP) | 0.42 | 0.46 | 0.50 | 0.27 | 0.23 | 0.25 |
| x8 (MLP) | 0.04 | 0.05 | 0.00 | 0.04 | 0.05 | 0.03 |

Table 10: Linear Probing $R^2$ Performance across the GIN layers for clustering and centrality measures (ClinTox dataset). Best Scores in Bold; Non-convergence indicated by —(full)

| GIN Layer | Clustering coef. | Transitivity | Assortativity | Avg. clustering | Avg. btw. cent. | PageRank cent. |
|-----------|------------------|--------------|---------------|-----------------|-----------------|----------------|
| x1 (GIN) | — | — | 0.32 | — | — | 0.18 |
| x2 (GIN) | — | — | 0.21 | — | — | — |
| x3 (GIN) | — | — | — | — | — | — |
| x4 (GIN) | — | — | — | — | — | — |
| x5 (GIN) | — | — | — | — | — | — |
| x_global | — | — | 0.25 | — | 0.48 | **0.40** |
| x6 (MLP) | — | — | **0.27** | — | 0.42 | 0.39 |
| x7 (MLP) | — | — | — | — | **0.47** | — |
| x8 (MLP) | — | — | — | — | 0.06 | — |

Table 11: Linear Probing $R^2$ Performance across the GIN layers for graph substructures (ClinTox dataset). Best Scores in Bold; Non-convergence indicated by —(full)

| GIN Layer | # Cliques | # Triangles | # Squares | Largest comp. size | Avg. degree | Graph energy |
|---|---|---|---|---|---|---|
| x1 (GIN) | 0.99 | — | 0.00 | 0.99 | 0.53 | **1.00** |
| x2 (GIN) | **1.00** | — | 0.00 | 0.99 | 0.46 | **1.00** |
| x3 (GIN) | 1.00 | — | 0.00 | **0.99** | 0.53 | 1.00 |
| x4 (GIN) | 0.99 | — | 0.00 | 0.99 | 0.20 | 0.99 |
| x5 (GIN) | 0.99 | — | 0.00 | 0.99 | — | 0.99 |
| x_global | 0.43 | — | 0.00 | 0.40 | **0.81** | 0.44 |
| x6 (MLP) | 0.43 | — | 0.00 | 0.40 | 0.80 | 0.44 |
| x7 (MLP) | 0.46 | — | 0.00 | 0.42 | 0.75 | 0.46 |
| x8 (MLP) | 0.04 | — | 0.00 | 0.04 | — | 0.05 |

Table 12: Linear Probing $R^2$ Performance across the GIN layers for spectral and small-world properties (ClinTox dataset). Best Scores in Bold; Non-convergence indicated by —(full)

| GIN Layer | Spectral rad. | Algebraic co. | Small world coef. | Small world idx | Avg. btw. cent. |
|---|---|---|---|---|---|
| x1 (GIN) | 0.70 | 0.78 | — | — | — |
| x2 (GIN) | 0.66 | **0.80** | — | — | — |
| x3 (GIN) | 0.61 | 0.80 | — | — | — |
| x4 (GIN) | 0.16 | 0.78 | — | — | — |
| x5 (GIN) | — | 0.69 | — | — | — |
| x_global | **0.74** | 0.67 | — | — | 0.48 |
| x6 (MLP) | 0.74 | 0.66 | — | — | 0.42 |
| x7 (MLP) | 0.71 | 0.56 | — | — | **0.47** |
| x8 (MLP) | 0.07 | 0.02 | — | — | 0.06 |

### E.2.2 PLOTS

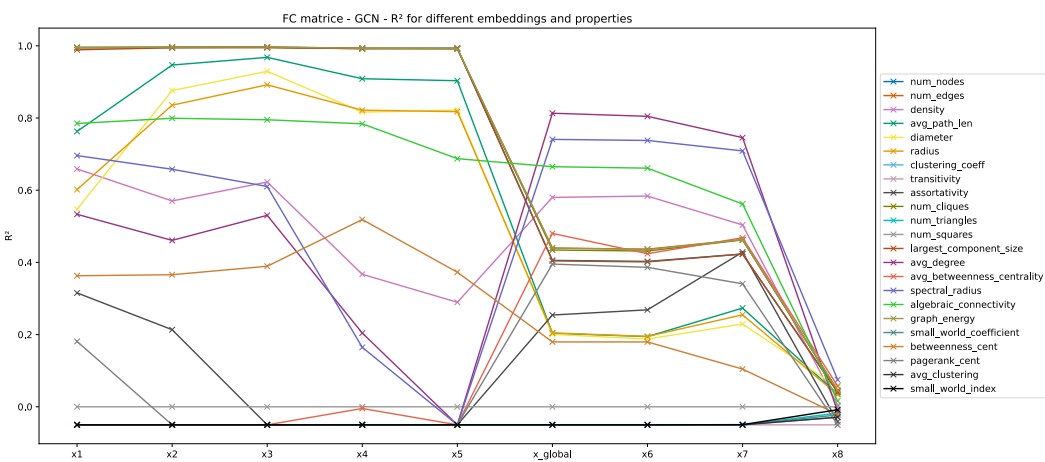

Figure 19: Plot of the GIN $R^2$ results across different layers probing for graph properties. ClinTox dataset (the negative $R^2$ values have been reduced to -0.05).

### E.2.3 Node properties probing results

Table 13: Linear Probing $R^2$ Performance across the GIN layers for various node properties (Clin-Tox dataset). Best Scores in Bold; Non-convergence indicated by —

| GIN Layer | degree | closeness | betweenness | eigenvector | clustering | pagerank |
|---|---|---|---|---|---|---|
| x0 (GIN) | **0.99** | 0.06 | 0.57 | 0.30 | — | 0.16 |
| x1 (GIN) | **0.85** | 0.12 | 0.51 | 0.31 | 0.00 | 0.20 |
| x2 (GIN) | **0.89** | 0.11 | 0.59 | 0.29 | — | 0.26 |
| x3 (GIN) | **0.86** | 0.07 | 0.51 | 0.28 | — | 0.17 |
| x4 (GIN) | **0.85** | 0.09 | 0.49 | 0.32 | — | 0.14 |

Here again, the very strong presence of the node degree makes a lot of sense when we know this property prepares the aggregation of global properties in the post pooling layers. The interesting thing is the non negligible presence of the betweenness centrality in all the layers which suggests that the betweenness centrality of atoms is important in the aggregation of global molecule properties that help predict the toxicity of a molecule. This property is more than the closeness or the clustering coefficient. The irreplaceable nature of some atoms in the molecular graph, which is literally the meaning of having a high betweenness centrality, is an important feature which makes these atoms targets to be part of higher order molecular schemes and patterns.

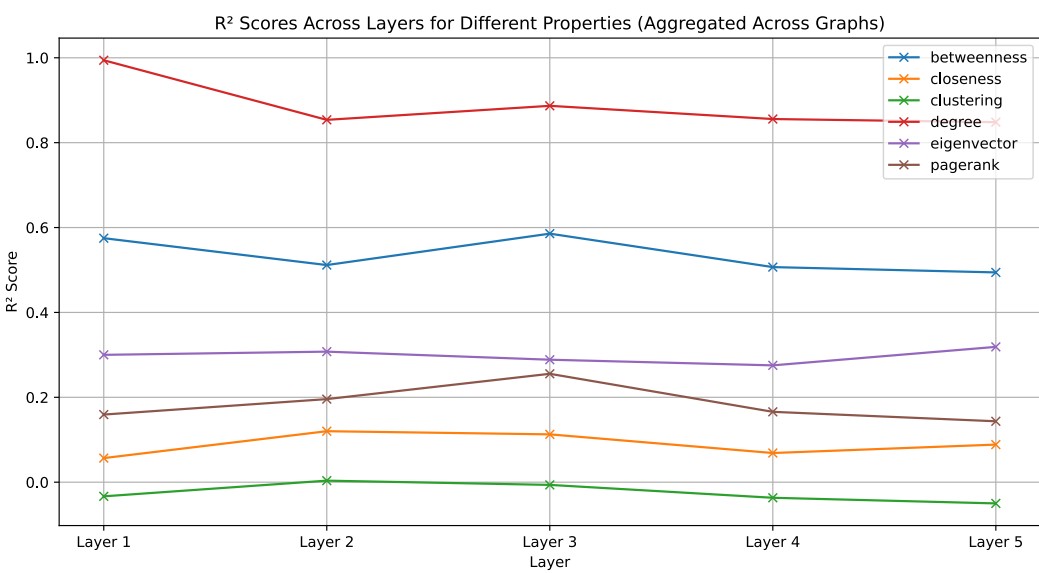

Figure 20: Plot of the GIN $R^2$ results across different layers probing for node properties. ClinTox dataset (the negative $R^2$ values have been reduced to -0.05). (full results)

# F  FMRI DATASETS

## F.1  MODELS

Table 14: Performance of Different Models on REST-meta-MDD and ABIDE with a 95%-5% Random Split. The highest performance is highlighted with boldface. All the performance of methods are reported under their best settings and round to the second decimal.

| Method | ABIDE | REST-meta-MDD |
|--------|-------|---------------|
| GCN | 0.56 | 0.61 |
| GIN | 0.69 | 0.69 |
| GAT | 0.62 | 0.67 |

Table 15: Range of Hyper-parameters and Final Specification for FC datasets

| Hyper-parameter | Range Examined | Final Specification |
|-----------------|----------------|---------------------|
| Graph Encoder | | |
| #GNN Layers | $\{[4, 5, 6]\}$ | 5 |
| #GIN Layers | $\{[4, 5, 6]\}$ | 5 |
| #GAT Layers | $\{[4, 5, 6]\}$ | 5 |
| #MLP Layers (for all models) | $\{[2, 3, 4]\}$ | 2 |
| #GCN Hidden Dimensions | $\{[64, 128, 256]\}$ | 128 |
| #GIN Hidden Dimensions | $\{[64, 128, 256]\}$ | 128 |
| #GAT Hidden Dimensions | $\{[64, 128, 256]\}$ | 128 |
| #GCN aggregation method | $\{[mean, sum, max(pooling)]\}$ | max pooling |
| #GIN aggregation method | $\{[mean, sum, max(pooling)]\}$ | mean pooling |
| #GAT aggregation method | $\{[mean, sum, max(pooling)]\}$ | max pooling |
| GCN Learning Rate | $\{[1e-2, 1e-3, 5e-4, 1e-4]\}$ | $5e-4$ |
| GIN Learning Rate | $\{[1e-2, 1e-3, 5e-4, 1e-4]\}$ | $5e-4$ |
| GAT Learning Rate | $\{[1e-2, 1e-3, 5e-4, 1e-4]\}$ | $1e-2$ |
| Batch Size (all models) | $\{32, 64, 128\}$ | 32 |
| Weight Decay (alll models) | $\{[1e-3, 1e-4]\}$ | $1e-4$ |
| batch normalisation | $\{with, without\}$ | $without$ |
| dropout | $\{with, without\}$ | $without$ |

## F.2 RESULTS ABIDE (ASD) DATASET

Table 16: Linear Probing $R^2$ Performance across GNN layers for basic graph properties (ASD dataset). Best Scores in Bold; Non-convergence indicated by —(full)

| GCN Layer | # Nodes | # Edges | Density | Avg. Path Length | Diameter | Radius |
|---|---|---|---|---|---|---|
| x1 (GCN) | — | **0.90** | — | 0.21 | 0.13 | 0.07 |
| x2 (GCN) | — | 0.77 | — | 0.22 | 0.24 | — |
| x3 (GCN) | — | 0.62 | — | — | 0.31 | — |
| x4 (GCN) | — | 0.38 | — | — | 0.14 | — |
| x5 (GCN) | — | 0.02 | — | — | 0.09 | — |
| x_global | — | 0.58 | 0.56 | 0.48 | 0.36 | 0.37 |
| x6 (MLP) | — | 0.52 | 0.50 | 0.45 | 0.39 | 0.41 |
| x7 (MLP) | — | — | — | — | — | — |
| **GIN Layer** | | | | | | |
| x1 (GIN) | — | 0.94 | — | 0.41 | 0.47 | 0.45 |
| x2 (GIN) | — | 0.55 | — | 0.38 | 0.28 | 0.23 |
| x3 (GIN) | — | 0.25 | — | 0.25 | — | — |
| x4 (GIN) | — | — | — | — | — | — |
| x5 (GIN) | — | 0.18 | — | — | — | — |
| x_global | — | 0.56 | 0.58 | 0.11 | 0.07 | 0.00 |
| x6 (MLP) | — | 0.58 | 0.66 | 0.14 | 0.10 | 0.09 |
| x7 (MLP) | — | 0.36 | 0.37 | 0.09 | 0.11 | — |
| x8 (MLP) | — | — | — | — | — | — |
| **GAT Layer** | | | | | | |
| x (GAT) | — | **0.93** | — | — | 0.16 | 0.04 |
| x2 (GAT) | — | **0.89** | — | 0.16 | 0.34 | 0.29 |
| x3 (GAT) | — | **0.84** | — | 0.30 | 0.39 | 0.31 |
| x4 (GAT) | — | **0.78** | — | 0.27 | 0.48 | 0.08 |
| x5 (GAT) | — | 0.67 | — | 0.52 | 0.44 | — |
| x_global | — | **0.74** | 0.70 | 0.60 | 0.29 | 0.40 |
| x6 (GAT) | — | **0.82** | 0.81 | 0.56 | 0.46 | 0.48 |
| x7 (GAT) | — | — | — | — | — | — |

Table 17: Linear probing performance ($R^2$ score) across GCN layers for clustering and centrality measures (ASD dataset). Best Scores in Bold; Non-convergence indicated by —(full)

| GCN Layer | Clustering coe. | Transitivity | Assortativity | Avg. clustering | Avg. btw. cent. | PageRank cent. |
|---|---|---|---|---|---|---|
| x1 (GCN) | — | — | — | — | — | — |
| x2 (GCN) | — | — | — | — | — | — |
| x3 (GCN) | — | — | — | — | — | — |
| x4 (GCN) | — | — | — | — | — | — |
| x5 (GCN) | — | — | — | — | — | — |
| x_global | 0.48 | 0.52 | 0.05 | 0.48 | 0.45 | 0.14 |
| x6 (MLP) | 0.33 | 0.30 | — | 0.33 | 0.41 | 0.06 |
| x7 (MLP) | — | — | — | — | — | — |
| **GIN Layer** | | | | | | |
| x1 (GIN) | — | — | — | — | — | — |
| x2 (GIN) | — | — | — | — | — | — |
| x3 (GIN) | — | — | — | — | — | — |
| x4 (GIN) | — | — | — | — | — | — |
| x5 (GIN) | — | — | — | — | — | — |
| x_global | 0.19 | 0.04 | — | 0.19 | 0.12 | — |
| x6 (MLP) | 0.23 | 0.08 | — | 0.23 | — | — |
| x7 (MLP) | 0.04 | — | — | 0.09 | 0.11 | — |
| x8 (MLP) | — | — | — | — | — | — |
| **GAT Layer** | | | | | | |
| x (GAT) | — | — | — | — | — | — |
| x2 (GAT) | — | — | — | — | — | — |
| x3 (GAT) | — | 0.02 | — | — | — | 0.02 |
| x4 (GAT) | — | — | — | — | — | — |
| x5 (GAT) | — | — | — | — | — | — |
| x_global | 0.44 | 0.08 | — | 0.41 | — | — |
| x6 (GAT) | 0.53 | 0.49 | 0.01 | 0.53 | — | 0.08 |
| x7 (GAT) | — | — | 0.00 | — | 0.00 | — |

Table 18: Linear probing performance ($R^2$ score) across GCN layers for graph substructures (ASD dataset). Best Scores in Bold; Non-convergence indicated by —(full)

| GCN Layer | # Cliques | # Triangles | # Squares | Largest comp. size | Avg. degree | Graph energy |
|---|---|---|---|---|---|---|
| x1 (GCN) | 0.51 | 0.88 | 0.54 | — | 0.85 | **0.90** |
| x2 (GCN) | 0.27 | **0.81** | 0.58 | — | 0.77 | 0.77 |
| x3 (GCN) | 0.06 | **0.73** | 0.40 | — | 0.64 | 0.62 |
| x4 (GCN) | — | **0.64** | 0.11 | — | 0.30 | 0.39 |
| x5 (GCN) | — | **0.61** | — | — | — | 0.04 |
| x_global | 0.46 | 0.42 | **0.61** | 0.19 | 0.57 | 0.58 |
| x6 (MLP) | 0.42 | 0.34 | 0.35 | 0.31 | 0.51 | **0.52** |
| x7 (MLP) | — | **0.00** | — | — | — | — |
| **GIN Layer** | | | | | | |
| x1 (GIN) | 0.58 | **0.95** | 0.69 | — | 0.94 | 0.95 |
| x2 (GIN) | — | **0.91** | 0.12 | — | 0.64 | 0.56 |
| x3 (GIN) | — | **0.74** | — | — | 0.22 | 0.25 |
| x4 (GIN) | — | **0.54** | — | — | — | — |
| x5 (GIN) | — | **0.75** | — | — | 0.23 | 0.17 |
| x_global | — | **0.86** | 0.14 | — | 0.57 | 0.56 |
| x6 (MLP) | — | **0.86** | 0.12 | — | 0.54 | 0.60 |
| x7 (MLP) | — | **0.59** | 0.00 | — | 0.37 | 0.36 |
| x8 (MLP) | — | — | — | — | — | — |
| **GAT Layer** | | | | | | |
| x (GAT) | 0.58 | 0.86 | 0.66 | — | **0.93** | **0.93** |
| x2 (GAT) | 0.56 | 0.82 | 0.69 | — | 0.87 | **0.89** |
| x3 (GAT) | 0.54 | 0.80 | 0.70 | — | 0.80 | **0.84** |
| x4 (GAT) | 0.50 | 0.75 | 0.74 | — | 0.75 | **0.78** |
| x5 (GAT) | 0.24 | **0.72** | 0.59 | — | 0.71 | 0.67 |
| x_global | 0.32 | 0.56 | 0.40 | — | 0.73 | **0.74** |
| x6 (GAT) | 0.51 | 0.76 | 0.52 | 0.20 | 0.81 | **0.82** |
| x7 (GAT) | — | — | — | — | — | — |

Table 19: Linear probing performance ($R^2$ score) across GCN layers for spectral and small-world properties (ASD dataset). Best Scores in Bold; Non-convergence indicated by —(full)

| GCN Layer | Spectral rad. | Algebraic co. | Small world coe. | Small world idx | Avg. btw. cent. |
|---|---|---|---|---|---|
| x1 (GCN) | 0.72 | — | — | — | — |
| x2 (GCN) | 0.74 | — | — | — | — |
| x3 (GCN) | 0.56 | — | — | — | — |
| x4 (GCN) | 0.36 | — | — | — | — |
| x5 (GCN) | — | — | — | — | — |
| x_global | 0.46 | 0.43 | — | 0.48 | 0.45 |
| x6 (MLP) | 0.38 | 0.41 | — | 0.39 | 0.41 |
| x7 (MLP) | 0.00 | — | — | — | — |
| **GIN Layer** | | | | | |
| x1 (GIN) | 0.88 | — | — | — | — |
| x2 (GIN) | 0.43 | — | — | — | — |
| x3 (GIN) | 0.25 | — | — | — | — |
| x4 (GIN) | — | — | — | — | — |
| x5 (GIN) | 0.24 | — | — | — | — |
| x_global | 0.76 | — | — | 0.40 | 0.12 |
| x6 (MLP) | 0.74 | — | — | 0.41 | — |
| x7 (MLP) | 0.18 | — | — | 0.23 | 0.11 |
| x8 (MLP) | — | — | — | — | — |
| **GAT Layer** | | | | | |
| x (GAT) | 0.79 | — | — | — | — |
| x2 (GAT) | 0.77 | — | — | — | — |
| x3 (GAT) | 0.02 | — | 0.02 | — | — |
| x4 (GAT) | 0.64 | — | — | — | — |
| x5 (GAT) | 0.49 | — | 0.09 | — | — |
| x_global | 0.58 | 0.20 | — | 0.38 | 0.56 |
| x6 (GAT) | 0.74 | 0.56 | 0.16 | 0.62 | 0.54 |
| x7 (GAT) | — | — | — | — | 0.00 |

## F.3 RESULTS REST-META-MDD (MDD) DATASET

Table 20: Linear probing performance ($R^2$ score) across GNN layers for basic graph properties (MDD dataset). Best Scores in Bold; Non-convergence indicated by —(full)

| GCN Layer | # Nodes | # Edges | Density | Avg. Path Length | Diameter | Radius |
|---|---|---|---|---|---|---|
| x1 (GCN) | — | **0.90** | — | — | — | — |
| x2 (GCN) | — | 0.85 | — | — | — | — |
| x3 (GCN) | — | 0.71 | — | — | — | — |
| x4 (GCN) | — | 0.64 | — | — | — | — |
| x5 (GCN) | — | 0.03 | — | — | — | — |
| x_global | 0.63 | 0.76 | 0.70 | 0.47 | 0.32 | 0.29 |
| x6 (MLP) | 0.60 | 0.67 | 0.60 | 0.33 | 0.23 | 0.18 |
| x7 (MLP) | — | — | — | — | — | — |
| **GIN Layer** | | | | | | |
| x1 (GIN) | — | **0.85** | — | 0.50 | — | — |
| x2 (GIN) | — | 0.67 | — | — | — | — |
| x3 (GIN) | — | — | — | — | — | — |
| x4 (GIN) | — | — | — | — | — | — |
| x5 (GIN) | — | — | — | — | — | — |
| x_global | — | 0.55 | **0.89** | — | — | — |
| x6 (MLP) | — | 0.55 | 0.60 | — | — | — |
| x7 (MLP) | — | 0.74 | 0.77 | — | — | — |
| x8 (MLP) | — | — | — | — | — | — |
| **GAT Layer** | | | | | | |
| x (GAT) | — | **0.94** | — | — | — | 0.04 |
| x2 (GAT) | — | 0.91 | — | — | — | — |
| x3 (GAT) | — | 0.86 | — | — | — | — |
| x4 (GAT) | — | 0.84 | — | — | — | — |
| x5 (GAT) | — | 0.73 | — | 0.20 | 0.16 | — |
| x_global | 0.52 | 0.80 | 0.74 | 0.29 | — | — |
| x6 (GAT) | 0.62 | 0.76 | 0.69 | 0.43 | 0.18 | 0.26 |
| x7 (GAT) | — | — | — | — | — | — |

Table 21: Linear probing performance ($R^2$ score) across GCN layers for clustering and centrality measures (MDD dataset). Best Scores in Bold; Non-convergence indicated by —(full)

| GCN Layer | Clustering coe. | Transitivity | Assortativity | Avg. clustering | Avg. btw. cent. | PageRank cent. |
|---|---|---|---|---|---|---|
| x1 (GCN) | — | — | — | — | — | — |
| x2 (GCN) | — | — | — | — | — | — |
| x3 (GCN) | — | — | — | — | — | — |
| x4 (GCN) | — | — | — | — | — | — |
| x5 (GCN) | — | — | — | — | — | — |
| x_global | 0.42 | **0.34** | — | 0.42 | 0.33 | — |
| x6 (MLP) | 0.35 | 0.33 | — | 0.35 | 0.41 | 0.11 |
| x7 (MLP) | — | — | — | — | — | — |
| **GIN Layer** | | | | | | |
| x1 (GIN) | — | — | — | — | — | — |
| x2 (GIN) | — | — | — | — | — | — |
| x3 (GIN) | — | — | — | — | — | — |
| x4 (GIN) | — | — | — | — | — | — |
| x5 (GIN) | — | — | — | — | — | — |
| x_global | — | — | — | — | — | — |
| x6 (MLP) | 0.22 | — | — | 0.22 | — | — |
| x7 (MLP) | 0.43 | 0.33 | — | 0.43 | — | — |
| x8 (MLP) | — | — | — | — | — | — |
| **GAT Layer** | | | | | | |
| x (GAT) | — | — | — | — | — | — |
| x2 (GAT) | — | — | — | — | — | — |
| x3 (GAT) | — | — | — | — | — | 0.02 |
| x4 (GAT) | — | — | — | — | — | — |
| x5 (GAT) | — | — | — | — | — | — |
| x_global | 0.45 | 0.59 | — | 0.45 | — | 0.24 |
| x6 (GAT) | **0.53** | 0.44 | — | **0.53** | — | 0.16 |
| x7 (GAT) | — | — | — | — | — | — |

Table 22: Linear probing performance ($R^2$ score) across GCN layers for clustering and centrality measures (MDD dataset). Best Scores in Bold; Non-convergence indicated by —(full)

| GCN Layer | Clustering coe. | Transitivity | Assortativity | Avg. clustering | Avg. btw. cent. | PageRank cent. |
|---|---|---|---|---|---|---|
| x1 (GCN) | — | — | — | — | — | — |
| x2 (GCN) | — | — | — | — | — | — |
| x3 (GCN) | — | — | — | — | — | — |
| x4 (GCN) | — | — | — | — | — | — |
| x5 (GCN) | — | — | — | — | — | — |
| x_global | 0.42 | **0.34** | — | 0.42 | 0.33 | — |
| x6 (MLP) | 0.35 | 0.33 | — | 0.35 | 0.41 | 0.11 |
| x7 (MLP) | — | — | — | — | — | — |
| **GIN Layer** | | | | | | |
| x1 (GIN) | — | — | — | — | — | — |
| x2 (GIN) | — | — | — | — | — | — |
| x3 (GIN) | — | — | — | — | — | — |
| x4 (GIN) | — | — | — | — | — | — |
| x5 (GIN) | — | — | — | — | — | — |
| x_global | — | — | — | — | — | — |
| x6 (MLP) | 0.22 | — | — | 0.22 | — | — |
| x7 (MLP) | 0.43 | 0.33 | — | 0.43 | — | — |
| x8 (MLP) | — | — | — | — | — | — |
| **GAT Layer** | | | | | | |
| x (GAT) | — | — | — | — | — | — |
| x2 (GAT) | — | — | — | — | — | — |
| x3 (GAT) | — | — | — | — | — | 0.02 |
| x4 (GAT) | — | — | — | — | — | — |
| x5 (GAT) | — | — | — | — | — | — |
| x_global | 0.45 | 0.59 | — | 0.45 | — | 0.24 |
| x6 (GAT) | **0.53** | 0.44 | — | **0.53** | — | 0.16 |
| x7 (GAT) | — | — | — | — | — | — |

Table 23: Linear probing performance ($R^2$ score) across GNN layers for graph substructures (MDD dataset). Best Scores in Bold; Non-convergence indicated by —(full)

| GCN Layer | # Cliques | # Triangles | # Squares | Largest comp. size | Avg. degree | Graph energy |
|-----------|-----------|-------------|-----------|--------------------|-------------|--------------|
| x1 (GCN)  | 0.52      | **0.77**    | 0.57      | —                  | 0.88        | **0.90**     |
| x2 (GCN)  | 0.58      | **0.84**    | 0.69      | —                  | 0.83        | 0.85         |
| x3 (GCN)  | 0.26      | **0.80**    | 0.55      | —                  | 0.72        | 0.72         |
| x4 (GCN)  | 0.04      | **0.79**    | 0.51      | —                  | 0.52        | 0.64         |
| x5 (GCN)  | —         | **0.52**    | —         | —                  | 0.01        | 0.04         |
| x_global  | 0.54      | **0.76**    | 0.50      | 0.62               | 0.73        | **0.76**     |
| x6 (MLP)  | 0.55      | **0.66**    | 0.44      | 0.62               | 0.63        | 0.67         |
| x7 (MLP)  | 0.06      | **0.10**    | —         | —                  | 0.08        | 0.09         |
| **GIN Layer** | | | | | | |
| x1 (GIN)  | 0.09      | **0.98**    | 0.58      | —                  | 0.86        | 0.85         |
| x2 (GIN)  | —         | **0.97**    | 0.45      | —                  | 0.48        | 0.67         |
| x3 (GIN)  | —         | **0.87**    | —         | —                  | 0.05        | —            |
| x4 (GIN)  | —         | **0.65**    | —         | —                  | —           | —            |
| x5 (GIN)  | —         | **0.22**    | —         | —                  | —           | —            |
| x_global  | —         | **0.91**    | —         | —                  | 0.70        | 0.58         |
| x6 (MLP)  | —         | **0.85**    | —         | —                  | 0.67        | 0.54         |
| x7 (MLP)  | 0.02      | **0.88**    | 0.51      | —                  | 0.75        | 0.74         |
| x8 (MLP)  | 0.02      | —           | —         | —                  | —           | —            |
| **GAT Layer** | | | | | | |
| x (GAT)   | 0.67      | **0.82**    | 0.70      | 0.07               | 0.93        | **0.94**     |
| x2 (GAT)  | 0.59      | **0.83**    | 0.81      | —                  | 0.89        | **0.91**     |
| x3 (GAT)  | 0.56      | **0.82**    | 0.83      | —                  | 0.82        | 0.86         |
| x4 (GAT)  | 0.51      | 0.87        | 0.79      | —                  | 0.82        | **0.84**     |
| x5 (GAT)  | 0.45      | **0.83**    | 0.26      | —                  | 0.67        | 0.74         |
| x_global  | 0.56      | **0.79**    | 0.68      | 0.56               | 0.78        | **0.80**     |
| x6 (GAT)  | 0.53      | **0.76**    | 0.55      | 0.65               | 0.73        | **0.76**     |
| x7 (GAT)  | —         | —           | —         | —                  | —           | —            |

Table 24: Linear probing performance ($R^2$ score) across GNN layers for spectral and small-world properties (MDD dataset). Best Scores in Bold; Non-convergence indicated by —(full)

| GCN Layer | Spectral rad. | Algebraic co. | Small world coe. | Small world idx | Avg. btw. cent. |
|---|---|---|---|---|---|
| x1 (GCN) | 0.52 | — | — | — | — |
| x2 (GCN) | 0.60 | — | 0.20 | — | — |
| x3 (GCN) | 0.53 | — | — | — | — |
| x4 (GCN) | 0.47 | — | — | — | — |
| x5 (GCN) | 0.07 | — | — | — | — |
| x_global | 0.60 | 0.63 | 0.28 | 0.31 | 0.33 |
| x6 (MLP) | 0.51 | 0.58 | 0.23 | 0.41 | 0.16 |
| x7 (MLP) | 0.04 | 0.04 | 0.00 | 0.00 | — |
| **GIN Layer** | | | | | |
| x1 (GIN) | 0.64 | — | — | — | — |
| x2 (GIN) | 0.52 | — | — | — | — |
| x3 (GIN) | 0.64 | — | — | — | — |
| x4 (GIN) | — | — | — | — | — |
| x5 (GIN) | — | — | — | — | — |
| x_global | 0.75 | 0.32 | 0.44 | 0.39 | — |
| x6 (MLP) | 0.73 | 0.20 | 0.43 | 0.40 | — |
| x7 (MLP) | 0.70 | 0.60 | 0.30 | 0.36 | — |
| x8 (MLP) | — | 0.03 | 0.01 | 0.00 | 0.01 |
| **GAT Layer** | | | | | |
| x (GAT) | 0.70 | 0.02 | 0.00 | — | — |
| x2 (GAT) | 0.66 | — | 0.23 | — | — |
| x3 (GAT) | 0.68 | — | 0.26 | — | — |
| x4 (GAT) | 0.73 | — | 0.30 | — | — |
| x5 (GAT) | 0.66 | — | 0.12 | — | — |
| x_global | 0.68 | 0.63 | 0.18 | 0.52 | 0.04 |
| x6 (GAT) | 0.63 | 0.59 | 0.21 | 0.50 | 0.25 |
| x7 (GAT) | — | — | 0.00 | — | — |

## F.4 PLOTS

**ASD**

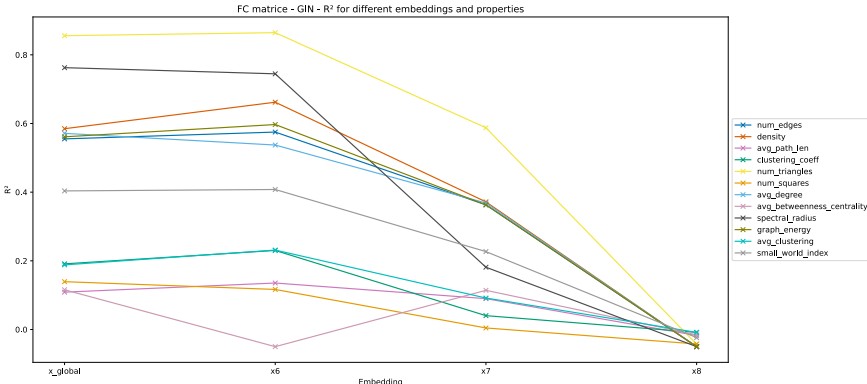

Figure 21: Plot of the GIN $R^2$ results across post pooling layers probing for graph properties ($R^2 < 0.1$ have been hidden). (ABIDE dataset)

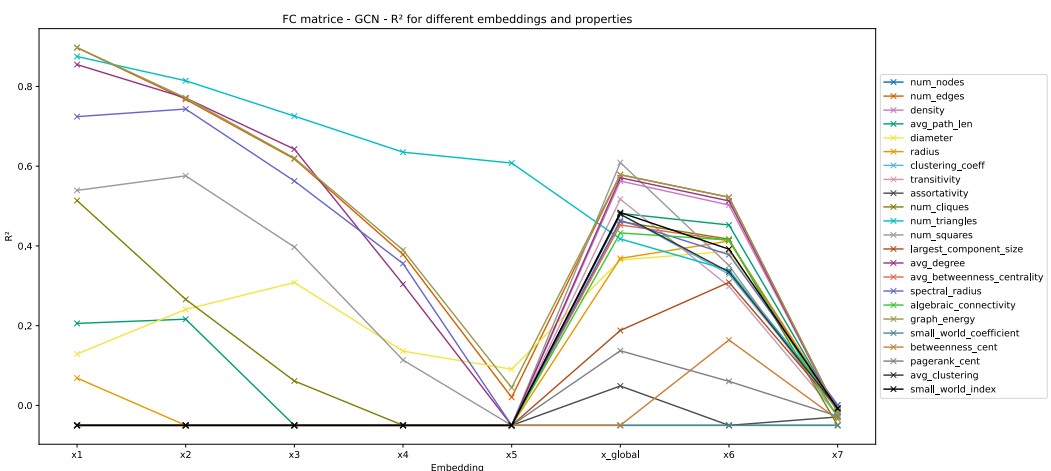

Figure 22: Plot of the GCN $R^2$ results across different layers probing for graph properties (ASD)

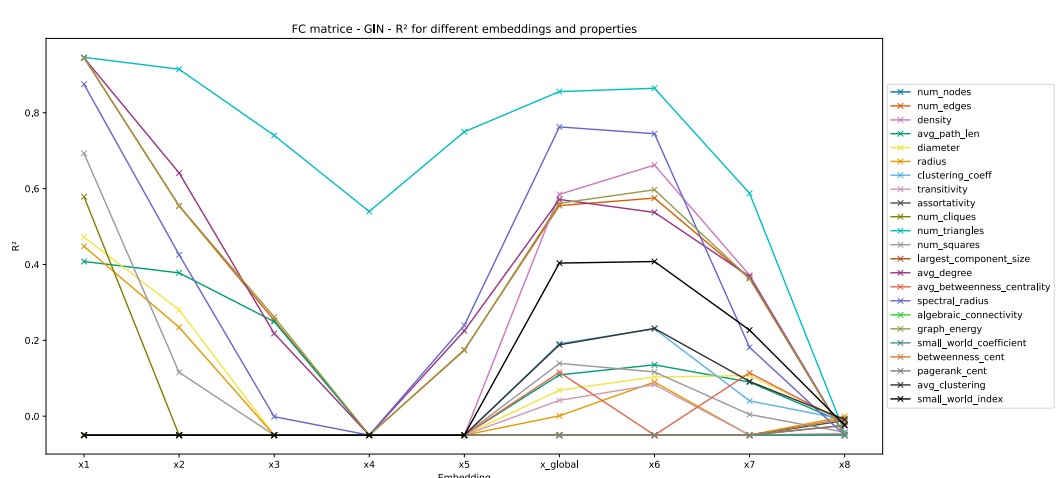

Figure 23: Plot of the GIN $R^2$ results across different layers probing for graph properties (ASD)

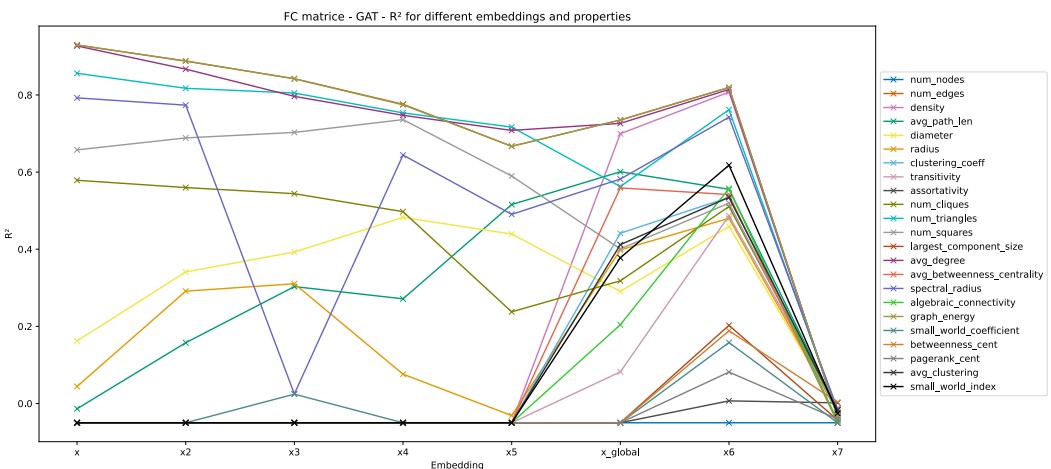

Figure 24: Plot of the GAT $R^2$ results across different layers probing for graph properties (ASD)

**MDD**

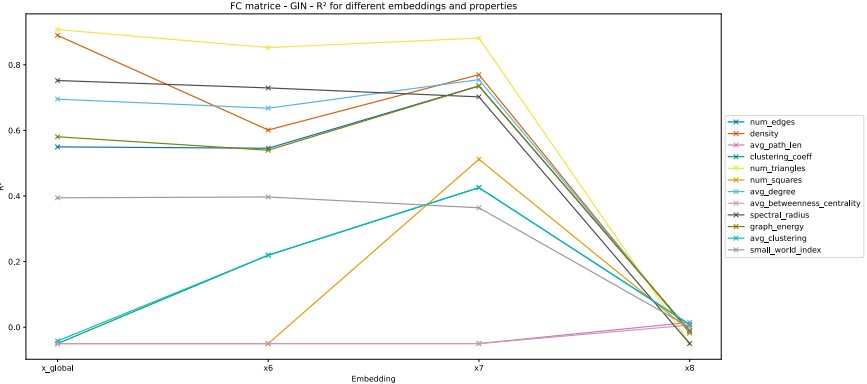

Figure 25: Plot of the GIN $R^2$ results across different layers probing for graph properties ($R^2 < 0.1$ have been hidden). (REST-meta-MDD dataset).

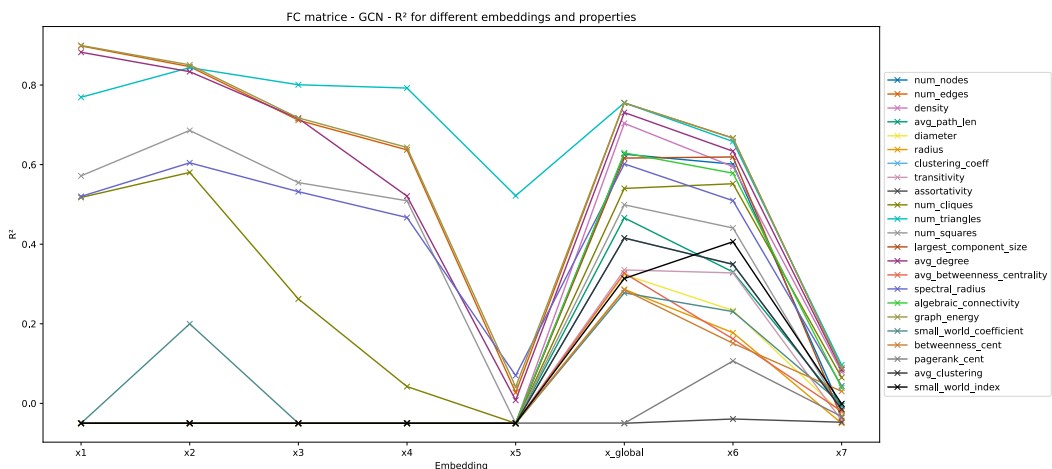

Figure 26: Plot of the GCN $R^2$ results across different layers probing for graph properties (MDD)

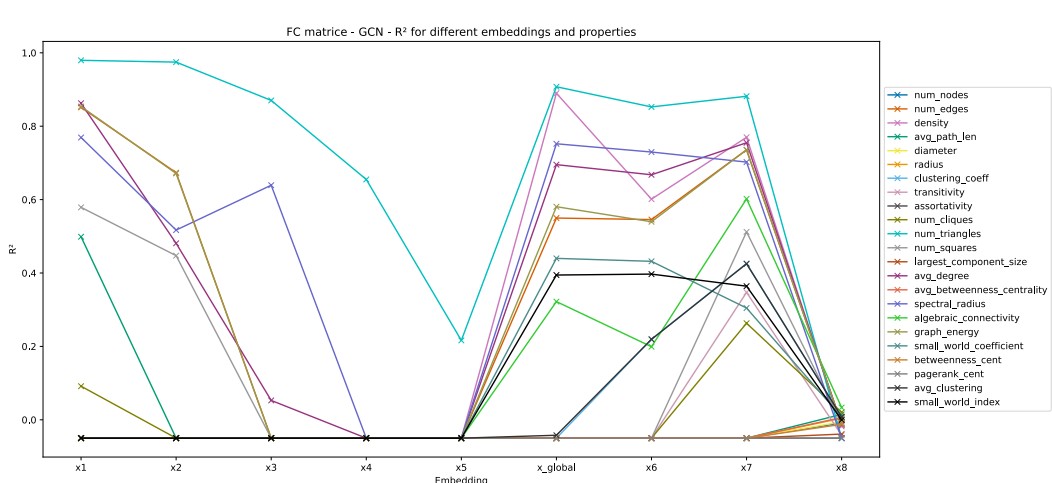

Figure 27: Plot of the GIN $R^2$ results across different layers probing for graph properties (MDD)

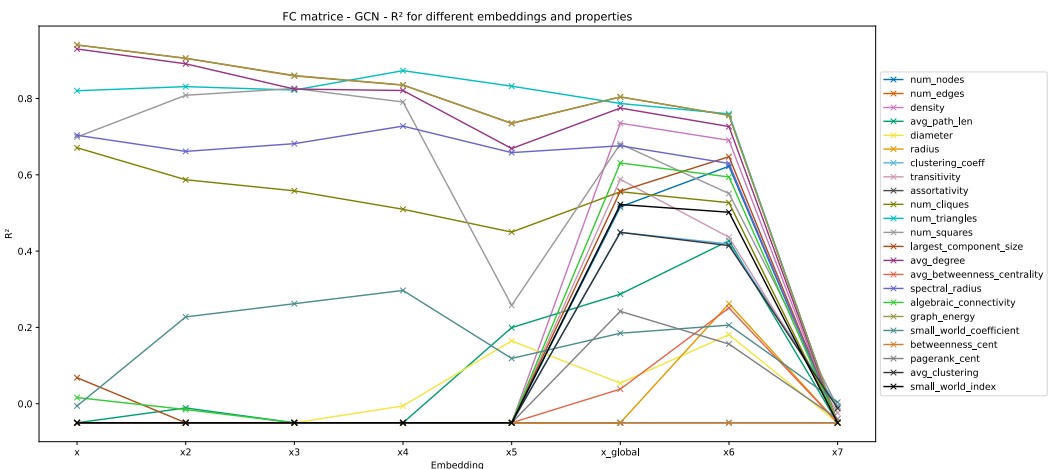

Figure 28: Plot of the GAT $R^2$ results across different layers probing for graph properties (MDD)

## F.5 RESULTS ASD AND MDD NODE PROPERTIES

Table 25: Linear probing performance ( $R^2$ score on the test set) across models for various node properties (ASD dataset). Best Scores in Bold; Non-convergence indicated by —

| GCN Layer | degree | closeness | betweenness | eigenvector | clustering | pagerank |
|---|---|---|---|---|---|---|
| x1 (GCN) | **0.83** | 0.26 | — | 0.37 | 0.12 | — |
| x2 (GCN) | **0.73** | 0.29 | 0.02 | 0.37 | 0.16 | 0.43 |
| x3 (GCN) | **0.61** | 0.23 | 0.02 | 0.35 | 0.17 | 0.40 |
| x4 (GCN) | **0.53** | 0.19 | 0.03 | 0.31 | 0.17 | — |
| out (GCN) | **0.53** | 0.20 | — | 0.27 | 0.16 | — |
| **GAT Layer** | degree | closeness | betweenness | eigenvector | clustering | pagerank |
| x1 (GAT) | **0.55** | 0.07 | 0.05 | 0.32 | 0.28 | 0.17 |
| x2 (GAT) | **0.52** | 0.48 | 0.08 | 0.31 | 0.30 | 0.14 |
| x3 (GAT) | 0.47 | **0.55** | — | 0.29 | 0.29 | — |
| x4 (GAT) | **0.41** | — | 0.14 | 0.19 | 0.26 | — |
| out (GAT) | 0.35 | **0.50** | 0.12 | 0.21 | 0.23 | — |
| **GIN Layer** | degree | closeness | betweenness | eigenvector | clustering | pagerank |
| x1 (GIN) | **0.90** | 0.38 | 0.05 | 0.42 | 0.14 | 0.57 |
| x2 (GIN) | **0.89** | 0.24 | 0.12 | 0.40 | 0.16 | 0.59 |
| x3 (GIN) | **0.80** | 0.35 | 0.12 | 0.38 | 0.13 | 0.51 |
| x4 (GIN) | **0.82** | 0.42 | 0.17 | 0.36 | 0.11 | 0.70 |
| out (GIN) | **0.83** | — | 0.13 | 0.30 | 0.13 | 0.70 |

For ASD results, the strong presence of Page Rank is interesting. Regardless of this, without surprise it's the degree that is consistently the highest node property as it prepare for global properties to aggregate.

Table 26: Linear probing performance ( $R^2$ score on the test set) across models for various node properties (MDD dataset). Best Scores in Bold; Non-convergence indicated by —

| GCN Layer | degree | closeness | betweenness | eigenvector | clustering | pagerank |
|---|---|---|---|---|---|---|
| Layer 0 | **0.83** | 0.30 | 0.05 | 0.38 | 0.16 | 0.40 |
| Layer 1 | **0.74** | 0.26 | 0.04 | 0.38 | 0.25 | — |
| Layer 2 | **0.69** | 0.31 | 0.03 | 0.41 | 0.23 | — |
| Layer 3 | **0.61** | 0.32 | 0.04 | 0.37 | 0.22 | — |
| Layer 4 | **0.61** | 0.33 | — | 0.37 | 0.19 | — |
| **GAT Layer** | degree | closeness | betweenness | eigenvector | clustering | pagerank |
| Layer 0 | **0.54** | 0.34 | — | 0.33 | **0.34** | 0.00 |
| Layer 1 | 0.55 | **0.60** | — | — | — | — |
| Layer 2 | **0.48** | 0.40 | — | 0.33 | 0.30 | 0.15 |
| Layer 3 | 0.43 | **0.65** | — | 0.29 | 0.28 | — |
| Layer 4 | **0.39** | — | — | 0.23 | 0.27 | — |
| **GIN Layer** | degree | closeness | betweenness | eigenvector | clustering | pagerank |
| Layer 0 | **0.92** | 0.54 | 0.09 | 0.40 | 0.23 | 0.58 |
| Layer 1 | **0.82** | 0.53 | 0.06 | 0.29 | 0.16 | 0.45 |
| Layer 2 | **0.83** | 0.43 | 0.16 | 0.34 | 0.18 | 0.60 |
| Layer 3 | **0.73** | 0.37 | 0.13 | 0.34 | 0.16 | 0.47 |
| Layer 4 | **0.86** | 0.24 | 0.20 | 0.26 | 0.11 | 0.47 |

The MDD dataset shows similar results which are surely explained by the same arguments.

# G  BRAIN IMAGING AND GNNS

Our brain is a network, more precisely a complex network of functionally interconnected regions specialised in specific cognitive tasks, sharing information with each other. In the last three decades, the field of biological neuroscience and computational cognitive neuroscience have provided and incredible amount of knowledge on the role, function and biological structure of such regions of interests, aiming at better understanding both the biological organisation of the brain (which we can refer to as the 'hardware implementation'), the representation embedded in this hardware and the computational strategy employed to treat this kind of representation (Marr, 1984). In other terms, we got better at understanding how each region independently organises itself and processes and forms information (cite Connecting network science and information theory). The main problem for modern computational neuroscience consists of understanding the brain's plasticity (how regions change over time), the inter-individual differences (how regions specialise differently between people) and how the brain integrates the information (how regions communicate with regard to each other).

For example we understand very well more basic brain structures like the cerebellum due to its high inter-individual similarity but we have a lot more difficulties modelling the prefrontal cortex which is so different from an individual to the other (Kanai & Rees, 2011; Gu & Kanai, 2014; Mills et al., 2021). In other terms, we do understand well the brain operating in segregation but not so much in integration (Aine, 1995). Functional segregation refers to the distinct specialisation of anatomical brain regions and functional integration refers to the possible temporal dependencies between the activity of anatomically separated regions of the brain.

Because the representation of a system composed by agents and interactions among them by a complex network is an effective way to extract information on the nature and topology of such interactions, it makes a lot of sense to study the integration of the brain network through its temporal dependencies. Understanding the mathematical properties of such a network with regard to some functional state of the brain network therefore helps understanding how the integration system of the brain and its architecture are linked to ways of processing information. Using Marr's paradigm to reformulate : understanding the functional communicative structure of the brain network helps understanding its algorithmic footprint. In terms of information theory, we could say that it helps understanding the relationship between topology and dynamics.

One way of accessing the brain activity is to use fMRI imaging. With fMRI measurements at ultra-high-field (3 Tesla, 7 Tesla or even 11 Tesla), hydrogen nuclei present in water and fat molecules align with the scanner's powerful magnetic field. When radio waves briefly disturb this alignment, the nuclei return to their initial alignment with the magnetic field, this is known as the resonance and causes local changes in the magnetic field. These changes are detected by receiver coils. The collected data from these interactions enable the precise determination of the 3D locations of these events, in the so-called voxels, which can then be visualised. This process underlies the BOLD (Blood Oxygen Level Dependent) response, which is crucial for functional Magnetic Resonance Imaging (fMRI) as it reflects changes in blood flow and oxygenation associated with neuronal activity. We use the magnetic response of blood flow as a proxy for brain activity.

Then, relying on fMRI, we have several ways to study the functional connectivity of the brain. Functional connectivity is defined as the temporal dependence of neuronal activity patterns of anatomically separated brain regions (Aertsen et al., 1989; Friston et al., 1993) and studies have shown that we could study functional connectivity between brain regions as the level of coactivation of functional MRI time-series (Lowe et al., 1998; 2000). As a result, conceptualising the brain as an integrative network of functionally interacting brain regions offers a powerful framework for understanding large-scale neuronal communication. It provides a method for investigating how functional connectivity and information integration relates to human behaviour and how this organisation may be altered in neurodegenerative diseases (Bullmore & Sporns, 2009; Greicius et al., 2009).

To understand how a specific brain region interacts with others, researchers most often analyse its resting-state activity and use simple pearson correlation of time-series data of a region with the time-series data of all other brain regions, they create a functional connectivity map (fcMap), which visually represents the strength of these connections (Biswal et al., 1997; Cordes et al., 2000). This is basically a matrix with value and we can understand it as a non relational data structure, in other terms, a graph.

More and more work in cognitive neurosciences explore the link between graph theory and connectomes (functional connectivity matrices) (Farahani et al., 2019). By representing brain regions as nodes and their connections as edges, graph theory provides a powerful framework for analysing the structural and functional organisation of the brain. Notably, studies have begun to explore the link between structural properties of brain connectivity, as captured by connectomes, and the manifestation of neurological disorders such as Autism Spectrum Disorders (ASD) and Major Depressive Disorders (MDD). ASD, characterised by impairments in social communication and repetitive behaviours. MDD is characterised as a mood disorder marked by persistent sadness and loss of interest.

These findings highlight the potential of connectome analysis to elucidate the neurological underpinnings of NDs and pave the way for the development of novel diagnostic and therapeutic strategies. Studying the link between the brain's Functional connectivity signature and behavioural quality of patients through probing learned embeddings of neural networks trained on classification tasks could thus be a promising avenue to help disentangle the gap between its segregational characteristics and the emergence (Johnson, 2002; Eccles, 1994; Wang et al., 2015; Carroll & Parola, 2024) of higher level behavioural quality.

However, if NDs result in alterations in brain functional and structural connections, as well as local and global connections (Seeley et al., 2009; Wang et al., 2015; Pasquini et al., 2015; Stam et al., 2007), traditional deep learning models such as CNN and LSTM are difficult to fit to the connectivity of the brain (Zhang et al., 2023). These long range dependencies, though, are well captured by the relational models defined previously in this thesis : Graph Neural Networks.

**Definition :** Psychiatric diagnosis can be regarded as a graph classification task. Given an input graph $\mathcal{G} = (\mathcal{V}, \mathcal{E})$ with node feature matrix $X$, GNNs employ the message-passing paradigm to propagate and aggregate the representations of information along edges to generate a node representation $h_v$ for each node $v \in \mathcal{V}$ and then explore the modelled human brains using graph methods to extract abnormal brain networks, subnetworks, and local connections (Palop et al., 2006; Thomas et al., 2016).

Similarly to (Zheng et al., 2023), a GNN can be formally defined through an aggregation function A and a combine function C such that $h_v^{(k)}$ is the node embedding of node $v$ at the $k$-th layer and $\mathcal{N}(v)$ is the set of neighbour nodes of $v$:

$$a_v^{(k)} = \mathrm{A}^{(k)} \left( \left\{ h_u^{(k-1)} : u \in \mathcal{N}(v) \right\} \right)$$
$$h_v^{(k)} = \mathrm{C}^{(k)} \left( h_v^{(k-1)}, a_v^{(k)} \right)$$

In the context of connectomes, many studies have focused on the relationship between general intellectual ability and small-world characteristics in intrinsic functional networks for describing individual differences in general intelligence (van den Heuvel & Hulshoff Pol, 2010; van den Heuvel et al., 2009; Langer et al., 2012; Hilger et al., 2017). Better intellectual performance was associated with shorter characteristic path length, the nodal centrality of hub regions in the salience network, as well as the efficiency of functional integration between the frontal and parietal areas (Jung & Haier, 2007) In general, when connections between specialised brain regions are disrupted, even within localised areas, the result is often functional impairment. This impairment is linked to atypical integration of activity across distributed brain networks (Ffytche & Catani, 2005; Catani et al., 2005). Characterising this impairment through the use of GNN could be one application of our probing pipeline. So far, GNNs have achieved promising diagnostic accuracy on autism spectrum disorder (ASD) (Rakhimberdina et al., 2020), schizophrenia (Rakhimberdina & Murata, 2020), bipolar disorder (BD) (Yang et al., 2019) and MDD (Zheng et al., 2023). We'll focus on ASD and MDD. But here as in other graph related fields, research has highlighted we were lacking Interpretability (Zheng et al., 2023).

For **ASD**, The contribution of rs-fMRI studies based on graph theory for autism exploration is important (Redcay et al., 2013; Rudie et al., 2013; Di Martino et al., 2014; Keown et al., 2017; Kazeminejad & Sotero, 2019). Studies have found increased short-range connections in ASD, particularly within sensory and association cortices. This local overconnectivity may contribute to the sensory

sensitivities and restricted interests often seen in ASD. Conversely, long-range connections between distant brain regions tend to be reduced in ASD. This underconnectivity affects integration of information across brain networks. Based on this literature (Farahani et al., 2019) we know that the modularity, *clustering coefficient*, and *local efficiency* are relatively reduced in ASD (i.e., inefficiency of information transmission in a particular module) while global communication efficiency is increased (shorter average path lengths). As another example, (Redcay et al., 2013) observed an increase in betweenness centrality and local connections by analysing the prefrontal brain areas in adolescents with ASD.

In the node property level, we would expect *betweenness centrality* to be one of the major properties linked with ASD. In the graph level level, we would thus expect the *clustering coefficient*, the degree to which connected nodes in the brain network are clustered together indicating increased local processing and functional segregation and over-connectivity in local brain regions. We would also expect the *characteristic path length* to be disrupted, the *average shortest path length* between all pairs of nodes in the network, suggesting differences in global information transfer efficiency. And *small-worldness* (SW) which quantifies the balance between local clustering and global integration. Atypical *SW* in ASD may reflect disrupted optimal network organisation imbalance between local and global processing. We would expect these properties to be critical in our GNNs embeddings trained on classification tasks.

For patients with **MDD**, several studies have reported topological changes in human brain connectome, including a loss of the *small-world network* (Ye et al., 2015; Achard & Bullmore, 2007)] and a significant reorganisation of the community structure (Zhang et al., 2011; Leistedt et al., 2009; Lord et al., 2012). In general, MDD patients exhibit increased *global and local clustering coefficients*, indicating a higher degree of local interconnectedness and efficiency in information processing. Moreover, increased *modularity* in MDD patients indicated that there were relatively less inter-modular edges and more intra-modular edges, which may also be associated with the disruptions in emotion regulation by decreasing communications between the Default Mode Network (DMN) and the Cognitive Control Network (CCN) (Ye et al., 2015). We would thus expect that classifying FC matrices with regard to MDD should use more *clustering coefficient*, *clusterization* properties and *modularity* measures than random (like the presence of motifs like the number small clusters, squares or triangles).

### G.1 Comparison between datasets

Comparing the ClinTox and the fMRI datasets an interesting observation emerges: basic graph properties (such as the *number of nodes*, of *edges* or the *average path length*) are almost omnipresent in the early layers of the GIN trained on the ClinTox dataset. However, their presence is less pronounced in the GIN trained on the ASD or MDD datasets. This difference offers a clue in distinguishing the complexity of brain-related neurological disorders from the complexity of chemical qualities such as toxicity. This suggests that the emergent properties of the brain may not be as easily tied to simple, differentiable structural features as those seen in molecular systems.

As a confirmation, the types of global graph properties present in the post pooling layers of the GIN-clintox model are of less high level of abstraction than the ones in GIN-MDD or GIN-ASD. The presence of the *average degree*, the *spectral radius*, the *algebraic connectivity* and the *density* as accurate explanations for the prediction of toxicity in molecules. The presence of the *spectral radius* in the last layer of the GIN makes it an even more interesting property to study for toxicity. On the other hand, the presence of motifs should be more investigated in the ASD and MDD datasets with eventually more complex motifs being probed (*hexagons* constituted of neighbored triangles, *house*, *grid*, etc).

