# OpenReview forum: "Do graph neural network states contain graph properties?"
_ICLR.cc/2025/Conference — Submitted to ICLR 2025_

### Official Review · Reviewer_5UPu · 2024-10-21

**Soundness:** 2
**Presentation:** 2
**Contribution:** 2
**Rating:** 3
**Confidence:** 4

**Summary:**

This paper claims that existing explainability methods are mostly instance-based and not well-suited for GNNs. It presents a model-agnostic explainability pipeline using diagnostic classifiers to better understand and trust GNN representations across various architectures and datasets. The findings are interesting and provide unique insights in the XAI for GNNs field. However, the paper has a confusing organization.

**Strengths:**

1. This paper discusses explainability from the model-agnostic aspect and explores the effectiveness of local graph properties.
2. This paper provides a comprehensive set of experiments on the local and global properties.

**Weaknesses:**

1. This paper is not well-organized. For example, in the introduction, the findings contain some figures and tables, but they can't be found in the main body. Since they are highly relevant to the findings, it would be better to present the results in the main body. The $R^2$ score is not well explained in the main body. In the appendix, there are some descriptions about the $R^2$ score; however, the mathematical formulation is not clear.
 2. Some key points are not clear about the experiments. For example, in line 133, it is confusing when discussing 'defining specific properties.' How to explore different local properties is not clear.
3. The figures are not well presented. For example, in Figure 3, the figures in the first row are hard to understand due to unclear captions. Additionally, it is not clear what $x$, $x_2$, and $x_3$ represent.

**Questions:**

1. What is the mathematical definition of $R^2$?
2. How are the different local properties discussed in the experiments?

---

> ### Author Response · Authors · 2024-12-02
>
> Thank you for your thoughtful comments. They were insightful and contributed to strengthen the paper. We understood better the ambiguities, the misses, the formulation imprecision and the presentation arrangement. Most importantly, we recognise the lack of organisation, presentation consistency and integrity. We rework the mathematical definition of the R2, both in the main body and in the appendix. We rework the methodology, the definition of graph properties, the mathematical definition of the probe, the results and the discussion in order to show better the results. Figures have been reworked and redesigned with better font (fig 1, 2, 3) and better legend (all the probing results). The use of local properties have also been clarified, as their results provide interesting interpretations but are not what we focus on (mainly because of resource limitations and because their use would be for a node classification paradigm). Let's dive into a bullet point analysis of your questions.
> 1. We acknowledge the lack of organisation and presentation. We worked on it with rewriting the different sections, reworking the figures, reworking the graphs, labels and mathematical definitions. You can find a good example with the rework of the probing definition or the R2 definition. In general, this helped to have clearer hypotheses like in the methodology and to present the results in a more insightful way, ordering in a clearer way how we investigate, address, demonstrate and validate clear hypotheses. This can be shown with the methodology and results sections where the supervision signal has been discussed through a finer analysis of the architectures (GCN message passing, GIN MLP injective aggregation function, GAT self attention).
> 2. We agree that local properties have not been highlighted enough and decided to radically focus on graph properties in our analysis in order to keep more coherency. They could still play a role in further research, in node classification setting for example, or to validate some hypotheses about how some graph properties emerge from local properties.
> 3. Figures have been reworked.
>
> Question 1 : R2 have been redefined.
>
> Question 2: see the answer to weakness 2.

---

### Official Review · Reviewer_FNiX · 2024-10-22

**Soundness:** 2
**Presentation:** 2
**Contribution:** 1
**Rating:** 3
**Confidence:** 4

**Summary:**

The paper explores the types of topological information encoded in node embeddings when solving downstream graph classification tasks. To reveal this information, the authors apply linear probing classifiers to predict various graph properties. The inputs are node embeddings computed at different depths across all graphs in the training data. Some of these properties can be inferred after the global pooling stage and are closely tied to the downstream task.

**Strengths:**

Linear probing is a straightforward yet powerful method, demonstrating strong results across the tested datasets. Its simplicity makes it easy to implement, while still providing valuable insights into the graph properties encoded by the GNNs.

Layer-wise analysis holds significant potential for identifying patterns in GNNs, much like the successes achieved in CNNs. This approach allows for a deeper understanding of how GNNs progressively capture and abstract graph properties across different layers.

**Weaknesses:**

"We addressed this by sorting the embeddings in descending order and padding with zeros at the end". However, this solution for handling node features before the global pooling stage seems somewhat arbitrary. An immediate alternative approach could be to consider different pooling strategies and even their concatenation.

Furthermore, Probing Graph Representations (Akhondzadeh et al., 2023) conducted a similar probing study, arriving at nearly identical conclusions. As a result, the amount of novel information introduced by this paper is limited. The authors also fail to compare their probing performance against that of Akhondzadeh et al., 2023, which would have been valuable.

Several key details are missing, such as the performance of the models on the main task and the correlation between task accuracy and probing accuracy. Although the referenced tables are provided in an extensive appendix, the main paper should be more self-contained to allow readers to verify the claims and key points easily.

Additionally, the text in Figures 2 and 3 is difficult to read, which hinders clarity.

**Questions:**

How does this work differ from the study "Probing Graph Representations" (Akhondzadeh et al., 2023)? In particular, I would like to focus on the limitations of the previous approach and the new insights provided by this current study.

---

> ### Author Response · Authors · 2024-12-02
>
> Thank you for your thoughtful comments. They were very insightful and contributed to strengthen the paper. We understood better the ambiguities, the misses, the formulation imprecision and the presentation arrangement. It helped us working on the overall structure of the new version, starting with a different and more effective introduction which showcase the difference with the *Probing Graph Representations* (Akhondzadeh et al., 2023) paper while highlighting better the original contribution of the paper, before developing stronger analyses on the layer-wise probing method and out-coming results. We also completely reworked the node feature probing pipeline, adopting and arguing better on two different strategies. This lead to new insights. In general, the integrity of the main paper, its 'self-containness', and the relevance and integrity of the different formulas, figures and definitions across the whole paper were arranged thanks to your feedback. Let's dive in a bullet point analysis of your comments.
>
> 1. I understand that you are talking about *paragraph 239*. It has been rework to be clearer but substantially here's my answer. First of all, a first confusion comes from the fact that we didn't explain enough why probing for graph properties in the node embeddings is a problem.
>
> 	To probe pre-pooling layers for global graph properties, we would need to concatenate and flatten the node embeddings with dimensions (number of nodes, number of features). A key issue here is that the flattening process introduces potential problems due to the non-canonical ordering of nodes, as their order in the dataset is arbitrary. This disrupts permutation invariance, since the linear classifier applied after flattening is not inherently invariant to node permutations. Post-pooling layers are naturally invariant, as their features have the shape (1, number of features).
>
> 	There were several options considered. Training on all possible permutations, ordering with a smart invariant method, using aggregation pooling method. Training on all possible permutations is impractical due to its computational cost. A more feasible approach involves sorting the nodes before flattening, using a criterion such as lexicographic ordering or embedding norms. This would ensure permutation invariance and allow embeddings of different graph sizes to be handled consistently. We opted to sort based on embedding norms to probe pre-pooling GNN layers. Here’s why: **Sorting by Norm Preserves Permutation Invariance**: The sorting mechanism we use orders node embeddings based on their norm in descending order. This operation depends only on the inherent properties of the embeddings themselves, not on their original ordering in the graph. As such, it inherently respects permutation invariance because reordering the nodes does not affect their norms or the resulting sorted order.
>
> 	Initially, we were concerned that probing pooled GNN layers might result in losing important, rich information that the probe is designed to highlight. However, we realized that probing tensors could introduce biases, where the classifier infers relationships and patterns between nodes, leveraging contextual relationships implicit in their embeddings—analyzing information that may arise from juxtaposed node vectors. This contrasts with the model itself, which lacks direct access to such representations. For instance, a probe could count the number of nodes by mapping the number of vectors in a concatenated tensor to the graph's node count, a capability not inherent to the model. To address this, we decided to probe mean-pooled embeddings and compare them to the tensor-based method. This approach led to new insights, such as observing that basic properties are absent from the mean-pooled representations. It aligns with expectations that properties like the number of nodes or edges are not captured in a pooled, graph-level embedding. More importantly, it allowed us to better explain the strong presence of complex motifs in the first layers of the GIN and GAT models, which ties to their expressivity and the theoretical grounding of their supervision signals.
>
> 2. We acknowledge having cited *Probing Graph Representations* (Akhondzadeh et al., 2023) in the original version of our paper. While we recognize we could have better articulated the originality of our contribution, we disagree with the claim of "nearly identical conclusions" as their probing study differs fundamentally from ours, particularly in the nature of the properties probed. Consequently, performance comparisons are not applicable. We encourage a closer reading of both their work and our introduction.
>
> 3. model performance is now clearly stated, correlations are highlighted, and the main text is more self-contained in terms of references, formulas, figures, and results.
>
> 4. Figures have been significantly improved (Figures 1, 2, 3, and the appendix).

---

### Official Review · Reviewer_oucj · 2024-10-28

**Soundness:** 3
**Presentation:** 2
**Contribution:** 3
**Rating:** 5
**Confidence:** 4

**Summary:**

The paper investigates the explainability of GNNs by proposing a model-agnostic pipeline that uses diagnostic classifiers to probe the embeddings of GNNs for known graph properties. It addresses the challenge of interpreting GNNs due to their complex, non-Euclidean structure and presents findings on how different GNN architectures and regularization techniques affect the representation of graph properties. The study demonstrates that GNN embeddings can indeed reflect graph-theoretic and domain-specific properties, offering insights into the models' internal states and potentially enhancing trust in their predictions. The approach is validated through applications to toxicity and fMRI datasets, confirming the alignment of probed properties with domain knowledge and uncovering previously unexplored structural properties.

**Strengths:**

1. The investigation tackles a problem that is both intriguing and significant. To the best of the reviewer's knowledge, this is the inaugural paper to address it.

2. The proposed pipeline is streamlined yet yields effective results.

3. The experimental outcomes provide valuable insights that will benefit future research endeavors.

**Weaknesses:**

This paper has several weaknesses. Addressing Weakness 2 & 3 & 4 will make a compelling case for 'accept.' Specifically:

1. The section designated for findings focuses more on articulating the experimental setting rather than the findings themselves. A clearer exposition of the results will fortify this section. (Question 1)

2. The use of GNNs, which are based on message passing and 1-WL, suggests that the inclusion of stronger GNNs could substantiate the experimental framework. (Question 2)

3. The authors have missed discussing the supervision signal, an important element that merits attention. (Question 3)

4. There is a lack of comprehensive explanations for the experimental results. Elaborating on these would enhance the paper's contribution to the field. (Question 4)

**Questions:**

1. The reviewer suggests that the findings section should encompass both the experimental settings and the derived conclusions. For instance, while the paper mentions an investigation into the effects of regularization techniques on learning, it fails to present the subsequent conclusions. A more comprehensive analysis, including both the setup and outcomes, would be beneficial.

2. Considering the expressiveness of GNNs, it is anticipated that stronger GNNs would more effectively capture graph properties. However, this may not hold true in practice due to generalization effects. It is recommended that the authors explore GNNs equivalent to 3-WL and consider examining graph transformers to bolster the paper's contributions.

3. Drawing parallels from computer vision research, where the supervision signal plays a pivotal role, it would be insightful to examine if self-supervised models in GNNs also capture structural information more effectively than supervised ones (like the results in CV).

4. While the extensive experiments contribute significantly to the field of Explainable AI (XAI), understanding the reasons behind varying performance levels is crucial. Questions such as why certain layers outperform others, why some substructures are more accurate, whether these findings correlate directly with the labels, and how the type of supervision influences the results, are all critical for a deeper comprehension of the study.

---

> ### Author Response · Authors · 2024-12-02
>
> Your acknowledgment of the pipeline's streamlined nature and its potential research applications is appreciated. The paper's strengths, highlighted through verbs like investigating, addressing, demonstrating, and validating, have helped refine the introduction effectively. I recognize the weaknesses you are listing, particularly the results's confused exposition and the confused presentation of the paper in general, the consistency of explanations across the paper and figures, the lack of analysis for the stronger architecture (like GIN and GAT architectures) and missing hypothesis formulation based on message passing and self-attention mathematical mechanisms. That was very insightful and important to note. Thank you for your thoughtful comments.
>
> Consequently, the mathematical definition of the probing was reworked, the methodology was formulated with clearer hypotheses, the supervision signal was discussed, the results were ordered differently with a better description and analysis of the observed results for the regularization methods, the comparison between models, the dataset specific results, the computer vision similarities. I believe these changes clearly helped to understand the reasons behind varying performance levels, highlight why certain properties performs better on some layers, why some substructures are more accurate and what is the role of the probing classifier itself in the interpretation of explanations. Some related changes has been done on the probing pipeline with a mean-pooling for GNN first layers. While the potential for enhancing contributions in 3-WL and unsupervised model probing was recognized, experimental investigations were constrained by time and resource limitations. I hope that this new version answer your interrogations, clarify some ambiguities and reinforce the strengths you've been pointing out. Let's dive in a bullet point analysis of your comments. Let's dive in a bullet point analysis of your comments.
>
> 1. A focus has been done on the structure : how hypotheses are formulated, how the setup is addressed, how we validate our hypotheses, what conclusions it drives and the possible new questions that the results rise. This is seen in the introduction, the methodology, the results and discussions sections.
>
> 2. In general, there is indeed room for 1) Expressing better the effect of the architecture on the graph properties embeddings. 2) Explore the role of GNNs equivalent to 1,2,3-WL and examine Graph transformer architecture. I worked on expressing better the effect of the architectures and what we know about their theoretical expressiveness. The message-passing mechanism in GNNs is designed such that each layer corresponds to one "hop" of neighborhood information: First layer → 1-hop neighborhood. Second layer → 2-hop neighborhood (as the aggregation is repeated). 𝑘 k-th layer → 𝑘 k-hop neighborhood. This is why in the first layer, the receptive field is limited to nodes directly connected to a given node. This lead to an interesting new outcome : GIN and GAT had, apparently, a good R2 score for the **number of square** in the first layer, which doesn't align with what we would expect. This lead to a few new hypotheses : If nodes in a square have unique **local features** or connectivity patterns, these could be partially encoded in the first-layer embeddings. More details are provided in the paper across the sections. With regard to the different models, we didn't have time and resource for training and probing the 3-WL architecture (which we coded and tried out before aborting). We hope that the new work and refined formulation would potentially strengthen the paper's contribution while still highlighting its own improvement possibilities in the further work section.
>
> 3. We've discussed the supervision signal while we could not have time to train and probe a self-supervised model. Same reasons as the part before.
>
> 4. In general, all the previous work built the foundation blocks to fully explain the difference between models, between properties and between layers. The supervision signal has been discussed through a finer analysis of the architectures (GCN message passing, GIN MLP injective aggregation function, GAT self attention).

---

### Official Review · Reviewer_Aw48 · 2024-11-04

**Soundness:** 2
**Presentation:** 1
**Contribution:** 2
**Rating:** 3
**Confidence:** 4

**Summary:**

This paper presents a model-agnostic approach to interpreting Graph Neural Network (GNN) embeddings by probing for encoded graph properties across various architectures and datasets. Using diagnostic classifiers, the study examines whether these embeddings represent structural properties such as clustering, density, and centrality. The method is tested on synthetic (Grid-House), molecular (ClinTox), and brain connectivity (fMRI) datasets, assessing each GNN’s capability to linearly separate embeddings based on these key properties.

**Strengths:**

1. Novelty: Proposes a new model-agnostic explainability approach using probing techniques for GNNs, which is less explored than instance-level explanations.

2. Comprehensive Dataset Testing: Validates the method across diverse datasets, showcasing its versatility.

3. Insightful Comparisons: Provides detailed comparative analysis of different GNN architectures (e.g., GCN, GAT, GIN), noting their strengths in capturing specific graph properties.

**Weaknesses:**

1. The padding and sorting steps for node embeddings, while intended to address non-uniform node counts, might introduce ordering biases and disrupt permutation invariance.

2. Some limitations in dataset representativity for real-world scenarios, as synthetic datasets like Grid-House may oversimplify the types of structures a GNN may encounter.

3. Fails to fully address the explainability of intermediate GNN layers, focusing more on final representations; probing deeper layers would provide more comprehensive insights into feature evolution.

4. This is my first time reviewing the ICLR paper, but I believe there is a large space to improve this manuscript, especially the presentation of the paper, which is the basic requirement for a good work.

**Questions:**

1. For the first sentence in the abstract — 'Graph learning models achieve state of the art performance on many tasks, but so at ever larger model sizes. Accordingly, the complexity of their representations increase.' — I found it hard to follow. As I understand it, this could be rephrased as: 'Graph learning models achieve state-of-the-art performance on many tasks, but this often requires increasingly larger model sizes, which in turn leads to more complex internal representations.'

2. For the images in Fig. 1, it would be better to improve the drawing styles and fonts. For example: the large black right bucket is better to be thiner and move down a little to vertically center alight with the GNN model and Dense layers.
Could the authors elaborate further on how their approach differs from other model-agnostic techniques, and clarify any specific novelty and contributions to the interpretability of intermediate layers?

---

> ### Author Response · Authors · 2024-12-02
>
> I acknowledge the novelty of the model-agnostic method, the versatility of applications and the insightful comparisons. The way you formulate these strength helped me reformulate the introduction in a more straightforward an to-the-point way. I recognize the weaknesses you are listing, particularly the ambiguity on the 'padding and sorting steps', the focus on later layers which is hard to understand in the way the paper was previously formulated and the Fig. 1 problems. That was very insightful and important to note. Thank you for your thoughtful comments.
>
> Consequently, I acted on two things, either I felt you were right and I changed things in the paper accordingly, either I interpreted the problems that you pointed out as a lack of clear communication and presentation from our part that would have create ambiguity. First, I changed the introduction to have a more straightforward approach, a more effective style and some more methodological explanations. I changed the fig. 1 to make it clearer and look more professional. I reworked the mathematical definition of probing to make it more consistent across the whole paper and make it clearer. I took into account the fact that using the norm of node vector embeddings was maybe not the best way to probe for graph properties in node representations while still arguing on why this technique is indeed permutation invariant and better than a simple concatenation. This lead to the exploration of a new probing technique with mean-pooled GNN layers, a clarification on what and why analyzing the results of the first layers is more interesting in a AI engineer point of view while analyzing the probing results of the last layers is more interesting from a domain-knowledge point of view. This also lead to a reorganization of the Methodology, the formulation of hypotheses and the results observed. I hope that this new version answer your interrogations, clarify some ambiguities and reinforce the strengths you've been pointing out. Let's dive in a bullet point analysis of your comments.
>
> 1. I understand that you are talking about *paragraph 239*. It has been rework to be clearer but substantially here's my answer. I acknowledge that probing for graph properties in the node embedding is strange. This lead to some conundrum for us. Concatenating without any ordering makes it not permutation invariant. Moreover, the probing classifier might learn to identify implicit patterns in the aggregated embedding as a weak signal. By operating on the set of embeddings simultaneously, it could **learn to infer relationships and patterns** between nodes, leveraging the **contextual relationships** implicit in their embeddings and 'seeing' properties that are not 'seen' as is by the model. For example knowing the number of nodes by knowing the number of vectors the concatenated tensor contain. Consequently, I tried to mean-pool the GNN layers in order to probe in a perfect permutation invariant setting. This also makes more sense with regard to graph properties being probe at a graph representation level. However, we loose the heursitic node embeddings. Here, **sorting by Norm Preserves Permutation Invariance**. The sorting mechanism we use orders node embeddings based on their norm in descending order. This operation depends only on the inherent properties of the embeddings themselves, not on their original ordering in the graph. As such, it inherently respects permutation invariance because reordering the nodes does not affect their norms or the resulting sorted order. If you meant that concatenation with zero-padding compromise invariance, we agree. This is precisely what we didn't do.
>
> 2. The Grid-House dataset, though admittedly synthetic and simplified, precisely for this reason, offers a controlled and well-defined environment for rigorously testing our hypotheses. This also have been better argued in the paper. we address the "laziness" phenomenon (Longa et al., 2023b) with controlled constraints while still having a ground truth explanation. This is ideal for a proof-of-concept experiments. They allow us to verify whether the GNN operates as intended in a setting where extraneous factors are minimized. If a model struggles with this dataset, it would likely underperform on real-world, more complex graphs, underscoring its diagnostic value.
>
> 3. I do agree that we were focusing on the analysis of later layers without explaining why. I first tried to explained better why this could be interesting from a domain-knowledge point of view while deepening the analysis of first layers as it was lacking a bit. I also developed more on the message passing paradigm and the expected behaviors of first layers and second layers. Each layer progressively expands the receptive field, mirroring how hierarchical feature learning works in CNNs for images.
>
> Questions 1: I reformulated the abstract, making it clearer, understanding your point.
>
> Question 2 : I changed fig 1, understanding your point.

---

### Meta-Review · Area_Chair_mHa8 · 2024-12-19

**Metareview:**

This paper proposes a model-agnostic pipeline for interpreting GNN embeddings by employing diagnostic classifiers to probe graph properties. The experiments demonstrate that the probed properties align well with domain knowledge.


However,all the reviewers tend to reject this paper and have pointed out some important weaknesses. First, the novelty of this paper is limited, as similar studies have already been conducted,  which arrive at comparable conclusions. Second, important details are missing. For example, some key points about the experiments are unclear, and there is a lack of comprehensive analysis of the results.


Therefore, I recommend the next version of this paper to incorporate more experiments and insights.

**Additional Comments On Reviewer Discussion:**

Reviewers Aw48, oucj, FNiX, 5UPu rated this paper as 3: reject, 5: borderline reject, 3: reject, and 3: reject, respectively.

All the reviewers tend to reject this paper and have pointed out some important weaknesses, including the limited novelty and insufficient experiments. Although the authors provided additional details and explanations, the main concerns raised by the reviewers, such as the lack of important experiments (e.g., probing performance experiment), have not been sufficiently addressed. Moreover, the reviewers did not provide positive feedback during the discussion phase. Therefore, I recommend the next version of this paper to incorporate more experiments and insights.

---

### Decision · Program_Chairs · 2025-01-22

Reject